# MIND THE GAP: EXAMINING THE SELF-IMPROVEMENT CAPABILITIES OF LARGE LANGUAGE MODELS

**Yuda Song**
CMU, Amazon
yudas@andrew.cmu.edu

**Hanlin Zhang**
Harvard University
hanlinzhang@g.harvard.edu

**Carson Eisenach**
Amazon
ceisen@amazon.com

**Sham M. Kakade**
Harvard University, Amazon
sham@seas.harvard.edu

**Dean Foster**
Amazon
foster@amazon.com

**Udaya Ghai**
Amazon
ughai@amazon.com

## ABSTRACT

Self-improvement is a mechanism in Large Language Model (LLM) pre-training, post-training and test-time inference. We explore a framework where the model verifies its own outputs, filters or reweights data based on this verification, and distills the filtered data. Despite several empirical successes, a fundamental understanding is still lacking. In this work, we initiate a comprehensive, modular and controlled study on LLM self-improvement. We provide a mathematical formulation for self-improvement, which is largely governed by a quantity which we formalize as the *generation-verification gap*. Through experiments with various model families and tasks, we discover a scaling phenomenon of self-improvement – a variant of the generation-verification gap scales monotonically with the model pre-training flops. We also examine when self-improvement is possible, an iterative self-improvement procedure, and ways to improve its performance. Our findings not only advance understanding of LLM self-improvement with practical implications, but also open numerous avenues for future research into its capabilities and boundaries.

## 1    INTRODUCTION

Recent work increasingly explores the use of synthetic data in training large language models (LLMs), with applications in both pre-training and post-training (Bai et al., 2022; Meng et al., 2022; Li et al., 2023; Adler et al., 2024; Dubey et al., 2024; Yang et al., 2024; Hui et al., 2024; Li et al., 2024). While synthetic data, often generated by LLMs, offers a valuable complement to human-generated data, its misuse can harm performance. Bertrand et al. (2023) and Gerstgrasser et al. (2024) showed self-training on model-generated data leads to degradation. To mitigate this, incorporating a "reliable" verifier to label data has shown promise in preventing such performance collapse (Gillman et al., 2024).

A straightforward verification mechanism is to train a reward model on human-annotated data to assess the quality of synthetic data (Lightman et al., 2023; Wang et al., 2024a). However, this approach can be prohibitively expensive and may offer few signals in domains where models exhibit super-human performance. An alternative is to use a stronger model (Chang et al., 2023; Havrilla et al., 2024) for annotation, but this becomes infeasible when the model is at the frontier of current capabilities. A promising solution is to use the model to label its own generations. Motivated by the intuition that "verification is easier than generation", one can hypothesize that the model may act as a better-than-random verifier of its own outputs, enabling *self-improvement* (Zelikman et al., 2022).

Most previous self-improvement algorithms can be summarized as follows: 1) make multiple generations from the model, 2) use the same model to verify the generations, and 3) distill from the reranked/filtered generation (Zelikman et al., 2022; Huang et al., 2022; Wang et al., 2022b; Yehudai et al., 2024; Madaan et al., 2024; Yuan et al., 2024; Xu et al., 2024; Liang et al., 2024). With this framework, self-improvement is also related to improving inference quality (Wang et al., 2022a; Welleck et al., 2024) – if the model can verify its own generation, self-improvement can enhance test-time performance with additional computation towards more generations and updates.

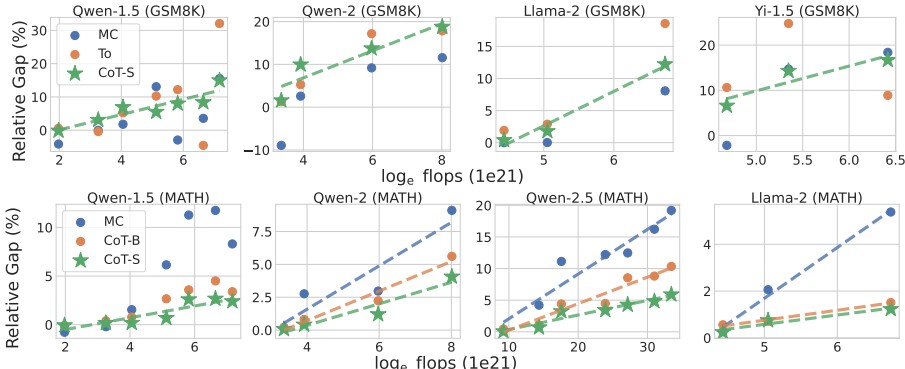

Figure 1: With proper verification method (e.g., CoT-S), *the relative generation-verification gap (Definition 2.2) scales monotonically with respect to the pre-training flops.* We conjecture that in this case, the relative gap is linear with respect to the log of the pre-training flops. MC denotes Multiple Choice verification, CoT-S denotes Chain-of-Thought-Score verification, and To denotes Tournament verification. The description of each verification can be found in Section 3.

Despite impressive empirical progress, a fundamental understanding of LLM self-improvement remains limited. It is uncertain whether these results can be interpreted solely as an indication of the self-improvement capability of LLMs, given the potential for confounders at various stages of the process . Moreover, much of the existing research focuses on just one model family or a single verification mechanism, limiting the broader applicability of the findings. In this work, we conduct a comprehensive study of the self-improvement capability of LLMs; our contribution is as follows: [1]

- **Self-Improvement Framework:** Section 2 details the mathematical formulation of the self-improvement process, highlighting three critical desiderata. We propose the *generation-verification gap (GV-Gap)* (Definition 2.1) as the central metric for evaluation. GV-Gap captures the "precision" of the model's verification over its own generations. Concretely, it is defined as the performance improvement obtained by re-weighting generations by the model's self-verification score (e.g., 0 or 1). Our empirical findings indicate that GV-Gap is a more accurate metric for measuring self-improvement versus the previous metric of the performance difference after a model update.

- **Scaling Properties:** In Section 4, we measure the generation-verification gap across multiple model families, verification mechanisms, and tasks. **i)** Certain verification methods induce a scaling phenomenon for self-improvement – *the relative GV-Gap (Definition 2.2) increases monotonically with pre-training flops* – shown in Figure 1. **ii)** We find that in cross-verification (i.e., using different models for generation and verification), the GV-gap increases with verifier capability and decreases with generator capability. **iii)** Finally, we observe that most models do not achieve self-improvement in information retrieval or reasoning tasks that exceed their inherent reasoning or planning capabilities. By studying multiple types of verification, our results indicate general patterns beyond just prompt engineering.

- **Iterative Self-Improvement:** In Section 5, we identify that **i)** GV-Gap saturates to 0 in handful rounds of iterative self-improvement; **ii)** the saturation rate is independent from the model capacity, **iii)** the effective diversity degrades during the iterative self-improvement.

- **Verification Mechanisms:** In Section 6, we consider methods to enhance self-improvement through a fine-grained study on the verification methods. Key observations include: **i)** the same verification method induces consistent trends among different models; **ii)** different verification mechanisms have significant non-overlaps; **iii)** GV-Gap is not necessarily positively correlated with generation accuracy; **iv)** an ensemble of verification can enhance self-improvement.

## 2 A DISSECTION OF THE SELF-IMPROVEMENT FRAMEWORK

In this section, we introduce the setup of the self-improvement framework considered in the paper, which consists of the following three main components:

---

[1]Due to space constraint we defer related works to Appendix B.

**Response Generation.** Let $\mathcal{X}$ be the prompt/context space, and $\mathcal{Y}$ be the response space. Let $\mathcal{F} \subseteq \{f : \mathcal{X} \to \Delta(\mathcal{Y})\}$ be a class of generative models that maps a prompt to a distribution over responses. A task, $\mathsf{task} \in \mathcal{T}$ (e.g., math, code, trivia, puzzles, etc.), defines a distribution $\mu_{\mathsf{task}} \in \Delta(\mathcal{X})$ over the prompt space $\mathcal{X}$, and utility function $u_{\mathsf{task}} : \mathcal{X} \times \mathcal{Y} \to [U_{\mathsf{min}}, U_{\mathsf{max}}]$, where $U_{\mathsf{min}}, U_{\mathsf{max}}$ denote bounds of the utility function. The goal is to find a generative model that maximizes the expected utility: $f^{\star}_{\mathsf{task}} := \arg\max_{f \in \mathcal{F}} J_{\mathsf{task}}(f)$, where

$$J_{\mathsf{task}}(f) := \mathbb{E}_{x \sim \mu_{\mathsf{task}}} \left[ \mathbb{E}_{y \sim f(\cdot|x)} \left[ u_{\mathsf{task}}(x, y) \right] \right].$$

We will often drop the subscript $\mathsf{task}$ when it is clear from the context.

**Verify with Proxy Utility.** Without the access to the ground truth utility function $u$, we rely on a proxy utility function constructed by some model $f$, denoted as $\widehat{u}_f : \mathcal{X} \times \mathcal{Y} \to [U_{\mathsf{min}}, U_{\mathsf{max}}]$. For example, let $Z = [10] := \{1, 2, \ldots 10\}$ be scores, the proxy utility $\widehat{u}_f(x, y) = \mathbb{E}_{z \sim f(\cdot|x,y,\mathsf{prom})}[z]$ rates the quality of the response with a score from 1 to 10 with an instruction prompt prom. In Section 3, we provide more proxy utility functions used in our experiments.

**Update via Reweighting.** Let $t \in [T]$ be the iteration index, $f_t$ be the model at iteration $t$, and $\widehat{u}_{f'}$ be the proxy utility defined by model $f'$. The reweighted distribution $f_t[w(\widehat{u}_{f'})]$ is defined such that

$$f_t[w(\widehat{u}_{f'})](y \mid x) \propto f_t(y \mid x) \cdot w(\widehat{u}_{f'}(x, y)), \forall x, y \in \mathcal{X} \times \mathcal{Y}.$$

Here $w : \mathbb{R} \to \mathbb{R}_{\geq 0}$ is a weight function that maps a utility from the verification procedure to a weight. The specific form of $w$ is determined by the algorithm used for the model update step (we provide two examples below). The objective is to find a model $f_{t+1} \in \mathcal{F}$ such that

$$\ell(f_{t+1}, f_t[w(\widehat{u}_{f'})]) := \mathbb{E}_{x \sim \mu}[d(f_{t+1}(\cdot \mid x), f_t[w(\widehat{u}_{f'})](\cdot \mid x))]$$

is small (i.e., $\ell$-projecting $f_t[w(\widehat{u}_{f'})]$ onto $\mathcal{F}$), for some distance measure $d$. Note that when $f' = f_t$, we have self-improvement, though the framework can also be used for improving using utilities produced by a different model as is studied in Section 4.2.

**Example 2.1** (KL-regularized RL Update). *One can treat the proxy utility $\widehat{u}$ as reward, and perform RLHF style RL update with a reverse KL constraint (Christiano et al., 2017; Ouyang et al., 2022):*

$$f_{t+1} = \arg\max_{f \in \mathcal{F}} \mathbb{E}_{x,y \sim \mu \circ f} \left[ \widehat{u}_{f_t}(x, y) - \beta \log\left( \frac{f(y \mid x)}{f_t(y \mid x)} \right) \right].$$

*In this case, we have $w(s) = \exp(s/\beta)$, and $d$ can be the KL divergence between $f_{t+1}(\cdot \mid x)$ and $f_t[w(\widehat{u}_{f_t})](\cdot \mid x)$ (Nemirovskij & Yudin, 1983).* ◁

**Example 2.2** (Rejection Sampling). *In rejection sampling, we first filter the generation by a threshold $\tau$, and then fine-tune the model on the filtered data:*

$$f_{t+1} = \arg\max_{f \in \mathcal{F}} \mathbb{E}_{x,y \sim \mu \circ f_t}[\log(f(y \mid x)) \cdot \mathbb{1}[\widehat{u}_{f_t}(x, y) \geq \tau]].$$

*In this case, we have $w(s) = \mathbb{1}[s \geq \tau]$, and $d$ can be the total variation distance between $f_{t+1}$ and $f_t[w(\widehat{u}_{f_t})]$ (Zhang, 2006).* ◁

Finally, it is convenient to abuse the notation and allow $w$ and $\widehat{u}$ to take batch input. For example, we can allow $w$ to take a list of score and then set the filtering threshold $\tau$ to the $n$ quantile ($n \in [0, 1]$) of the score. We denote this as top-$n$ or quantile-$n$ filtering.

## 2.1 Three Key Factors of Self-improvement

For any meaningful self-improvement, at iteration $t$, we would like to find $f_{t+1}$ such that $J(f_{t+1}) > J(f_t)$, where recall $J(f)$ is the expected utility under the model $f$. We identify the three key conditions that may bottleneck improvement on model $f$: **1. Improvable Generation.** Our framework involves reshaping the generation distribution towards increased utility. In order for this to be useful, the utilities of generations must have variability. For example, if generation were done with greedy decoding, no improvement in this process would be possible. Fortunately, the improvable generation phenomenon has been well-observed in LLMs (Li et al., 2022; Brown et al., 2024) (see also Figure 4). **2. Informative Verification.** Recall that weight function $w(\widehat{u}_g)$ is defined by the proxy utility function $\widehat{u}_g$, which is constructed by a verifier model $g$. If the verification capability is limited, the weighting may not provide a useful signal for improvement. The following definition quantifies this intuition:

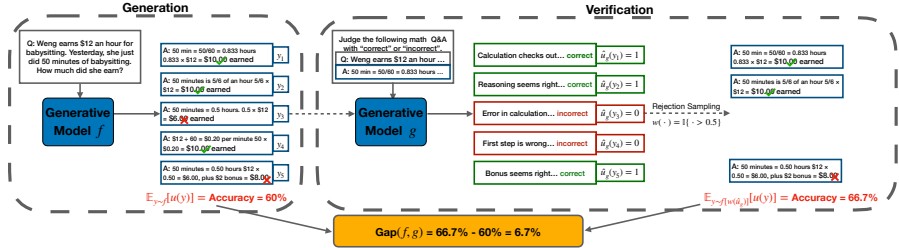

Figure 2: A visualization of using rejection sampling as an example for the key definitions in the self-improvement framework. For simplicity, we assume a single prompt $x$.

**Definition 2.1** (Generation-Verification Gap). *For a generator $f$ and verifier $g$, we define the generation-verification gap (GV-Gap) between $f$ and $g$ as*

$$\mathsf{gap}(f,g) := J(f[w(\widehat{u}_g)]) - J(f) .$$

We include a visualization of Definition 2.1 in Figure 2. This is the core metric for our analysis throughout the paper. When the generator $f$ and verifier $g$ are the same, we denote the shorthand $\mathsf{gap}(f) := \mathsf{gap}(f,f)$, which is the self-improvement generation-verification gap. However, the GV-Gap is an absolute quantity, which does not fully capture the various qualities of the generation. Consider a generator that already achieves 99% accuracy on a task: first, the upper bound for $\mathsf{gap}(f)$ is only 0.01; second, incorrect responses are likely to be very subtle, and thus any improvement in the reweighted distribution might require a very strong verification model. This motivates the relative GV-Gap:

**Definition 2.2** (Relative Generation Verification Gap). *For a generator $f$ and verifier $g$, we define the relative generation-verification gap between $f$ and $g$ as*

$$\mathsf{gap}_{\mathsf{rel}}(f,g) := \mathbb{E}_{x\sim\mu}\left[\frac{\mathbb{E}_{y\sim f[w(\widehat{u}_g)](\cdot|x)}[u(x,y)] - \mathbb{E}_{y\sim f(\cdot|x)}[u(x,y)]}{U_{\mathsf{max}} - \mathbb{E}_{y\sim f(\cdot|x)}[u(x,y)]}\right] ,$$

That is, we weigh the gap of each prompt by its deficiency to the best possible utility. For simplicity, we will denote the self-GV-Gap as "gap" or $\mathsf{gap}$ and relative self-GV-Gap as "relative gap" or $\mathsf{gap}_{\mathsf{rel}}$ when the context is clear. In domains where verification is easier than generation, $\mathsf{gap} > 0$ likely holds, and indicates that there is additional signal that can be exploited. One can also check that, for all prompts $x \in \mathcal{X}$, if the weight function $w(\widehat{u}_g)(x,\cdot)$ and $u(x,\cdot)$ is positively correlated[2] under the distribution of $y \sim f$, then we can always guarantee $\mathsf{gap}(f,g) > 0$.

**3. High-fidelity Model Update.** The final condition is that the model $f_{t+1}$ mimics/distills the performance of the reweighted distribution $f_t[w(\widehat{u}_{f_t})]$, i.e., $|J(f_{t+1}) - J(f_t[w(\widehat{u}_{f_t})])| \leq \varepsilon_{\mathsf{update}}$, with some small $\varepsilon_{\mathsf{update}}$. For example, if through MLE we bound the TV-distance between the two by $\varepsilon'$, then by Holder's inequality we have $\varepsilon_{\mathsf{update}} \leq \varepsilon' U_{\mathsf{max}}$. Combining it with the gap guarantee, we have:

$$J(f_{t+1}) - J(f_t) \geq \mathsf{gap}(f_t, g) - \varepsilon_{\mathsf{update}}.$$

With a sufficiently expressive LLM class, it is often observed that the distillation error $\varepsilon_{\mathsf{update}}$ is small.

Note that sometimes we might observe $J(f_{t+1}) - J(f_t[w(\widehat{u}_{f_t})]) > 0$. For instance, a benchmark may require outputs in a specific format; in such cases, the finetuned model $f_{t+1}$ might outperform the reweighted distribution $f_t[w(\widehat{u}_{f_t})]$ simply by aligning outputs with the required format, even if the underlying answers remain unchanged. Recent works (Dubois et al., 2024; Zhang et al., 2024c) highlight additional confounders, such as modifications to output length during finetuning, which may inflate perceived improvements without reflecting true model capabilities. Conversely, it is conceivable that intrinsic enhancements in the model's reasoning might occur; for example, mastering simpler tasks could indirectly boost performance on more complex problems requiring similar skills. However, in our experiment we only observe the former scenario. That said, both cases emphasize the need for caution when interpreting such improvements, and this further emphasizes the importance of our modular approach in dissecting the components of self-improvement.

---

[2]Even under the case that $w(\widehat{u}_g)(x,\cdot)$ and $u(x,\cdot)$ are negatively correlated, if we have a small holdout dataset with ground truth labels $u(x,y)$, we can define $w(\alpha) := \exp(\alpha \cdot w((\widehat{u}_g)))$, use the holdout set to tune $\alpha$, and use $w(\alpha)$ to reweight the distribution.

## 3 EXPERIMENT SETUP

Our experiment is based on lm-evaluation-harness (Gao et al., 2024). For all tasks we use the following setup: for generations and verification, we use sampling parameters $p = 0.9, t = 0.7$, max length of 512 and 4-shot in-context samples.[3] For each model $f$, for each prompt $x$, we sample 128 responses $y \sim f(x)$, and sample 1 verification for each response, which defines the proxy utility score $\hat{u}_f(x, y)$.[4] We mainly consider the rejection sampling setting (for completeness we investigate the RL setting in Section 5), and thus the weight function $w$ is the indicator function with either quantile or global threshold (c.r. Example 2.2). Then we calculate gap or $\text{gap}_{\text{rel}}$ according to Definitions 2.1 and 2.2, which is the accuracy difference between the filtered generations and the original generations. Specifically, we consider the following verification mechanisms (a formal description of each is provided in Appendix C along with example prompts in Appendix G):

1. **Multiple Choice (MC)** (Zhao et al., 2023; Liu et al., 2023; Dong et al., 2024) asks the LM to label responses as "Correct" and "Incorrect" and uses the probability of "Correct" as a continuous score.

2. **Chain of Thought (CoT)** asks the LM to score responses and to provide the justification (i.e. CoT (Wei et al., 2022)) and the score is parsed from the answer. Scores can be on a scale from 1 to 10 (**CoT-Score**) (Yuan et al., 2024; Liang et al., 2024) or binary (**CoT-Binary**).

3. **Tournament (To)** involves sampling a batch of generations and having a verifier compare generation pairs in a single elimination tournament to produce a new generation distribution. We can repeat the process until there is only one response left from the batch.

We consider the following models families: Qwen-1.5 (Bai et al., 2023), Qwen-2 (Yang et al., 2024), Qwen-2.5 (Team, 2024), Llama-2 (Touvron et al., 2023), Llama-3, Llama-3.1 (Dubey et al., 2024) and Yi-1.5 (Young et al., 2024). To avoid the confounding effect of the post-training, all experiments in this paper are performed on *base* models.[5] Finally, all inference in this paper is performed with vLLM (Kwon et al., 2023).

## 4 SCALING PROPERTIES OF GENERATION-VERIFICATION GAP

In this section, we conduct a comprehensive study on measuring the scaling property of the generation-verification gap due to its valuable practical guidance in both pre-training and downstream tasks (Kaplan et al., 2020; Hernandez et al., 2021; Isik et al., 2024; Ruan et al., 2024).

### 4.1 SCALING RESULTS

We start with the GSM8K benchmark (Cobbe et al., 2021), with 1320 questions on the test data split, and MATH benchmark (Hendrycks et al., 2021), with 5000 questions on the test data split. The ground truth utility $u(x, y) = 1$ if the end of response $y$ is the correct answer to the question $x$, or $u(x, y) = 0$ otherwise. We compute $\text{gap}(f)$ for each model $f$ and verification method, and we record the full results in Tables 3 and 4. In particular, we observe the following phenomena:

**Small Models can not Self-improve.** For small (in terms of pre-training flops) models such as Qwen-1.5 0.5B, Qwen-2 0.5B and Llama-2 7B, $\text{gap}(f)$ is non-positive for nearly all verification methods, even though the models have non-trivial generation accuracy. We also observe this phenomenon in Pythia (Biderman et al., 2023) and OPT (Zhang et al., 2022) model families. We believe this result indicates that self-improvement requires a minimal level of instruction following and reasoning capabilities, which is not present in these small models. We will further illustrate this point in Section 4.4.

**CoT Verification is More Stable than MC.** Some MC verification incurs non-positive gap even for medium-sized models such as Qwen-1.5 14/32B and Llama-3/3.1 8B models, while CoT verification

---

[3]In Appendix D.1, we perform ablation on different sampling temperatures, and observe that the same observations hold for a range of reasonable hyperparameters.

[4]Note that an ideal verification should be sampling multiple verifications per generation. We only sample one due to computational constraints and we leave multiple verifications along with understanding verification compute scaling for future work. As a preliminary investigation on this issue, in Appendix D.4, we perform small-scale experiments on measuring the variance of the verification in self-improvement.

[5]In Appendix D.2, we repeat a subset of our experiments on the instruct models and we indeed observe that the results are more noisy.

Figure 3: GV-Gaps (%) in cross-improvement. For each row (a fixed generator), gap increases as verifier capacity goes up. For each column (a fixed verifier), gap decreases as generator capacity goes up.

always has a positive gap for medium/large-sized models. Our results align with recent studies showing that MC evaluation might be unreliable, especially for small models (Dominguez-Olmedo et al., 2024). We perform a more in-depth analysis on this point in Section 6.

$\mathsf{gap_{rel}(f)}$ **Scales with Pre-training Flops.** We observe that with certain verification methods (such as CoT-Score), the relative gap grows monotonically with the pre-training flops, demonstrating a scaling property. We visualize the scaling results in Figure 1, where we plot $\mathsf{gap_{rel}}(f)$ with respect to the logarithm of pre-training flops. Specifically, we hypothesize that in the case where the verification elicits the scaling property, $\mathsf{gap_{rel}}(f)$ scales linearly with respect to the logarithm of the pre-training flops. However, note that we should not expect the slope for each model family to be the same. In Figure 12, we repeat the same plot for $\mathsf{gap}(f)$, but we do not observe a similar trend with the absolute gap.

## 4.2 CROSS VERIFICATION

In self-improvement, both generator and verifier change when transitioning between different models. To better understand the relationship between generation/verification ability and model capacity, we perform a cross-verification study, where we only alter either the generator or the verifier at a time. We consider the Llama-2 and Qwen-2 model families, and the two most representative verification methods: MC with quantile threshold and CoT-Score. We present the results in Figure 3. We observe that the results are consistent with our intuition on the difficulty of verification: fix a generator model $f$, $\mathsf{gap}(f, g)$ increases as the model capacity (defined by pre-training flops) of the verifier model $g$ increases. On the other hand, fix a verifier model $g$, $\mathsf{gap}(f, g)$ decreases as the model capacity of the generator model $f$ increases, as the error of the generator model becomes more difficult to detect.

At first glance, the results seem to imply that selecting the largest model as the verifier, akin to a teacher-student setup, is advantageous. However, considering the computational costs associated with larger verifier models, this approach might be suboptimal. Alternatively, a weak-to-strong setup, where a smaller model verifies a larger one, might be more cost-effective, but our findings indicate that a positive gap cannot always be assured. We believe an interesting future direction is to explore the compute-optimal configuration for cross-verification. This, however, might require a combinatorially large number of experiments to pinpoint the optimal verifier for each generator.

> **Takeaway on scaling of self-improvement**
>
> LLMs demonstrate clear scaling trends in self- and cross-improvement:
>
> - **Self-Improvement:** With stable verification, the relative gap increases monotonically with pre-training flops.
> - **Cross-Improvement:** The gap scales **directly** with the verifier's flops and **inversely** with the generator's flops.
>
> If the relative gap scales linearly with the logarithm of pre-training flops, this relationship could guide decisions on synthetic data generation strategies in self-improvement. Additionally, results from cross-verification suggest that a compute-optimal combination may exist to maximize efficiency in cross-improvement contexts.

## 4.3 UNIMPROVABLE TASKS

The primary objective of self-improvement is predicated on the assumption that "verification is easier than generation". As such, it is also worthwhile to consider tasks where such intuition would not

Table 1: Gap (%) on Natural Question for Qwen-2 models. While all models have a non-trivial generation accuracy, all gaps are near 0, indicating that the task is unimprovable.

|  | 0.5B | 1.5B | 7B | 72B |
|---|---|---|---|---|
| Generation Accuracy | 6.51 | 13.87 | 29.09 | 41.45 |
| MC (top 0.8) | -0.06 | 0.04 | 0.79 | 0.28 |
| MC ($\tau = 0.8$) | -0.05 | 0.02 | -0.05 | -0.05 |

Table 2: Generation accuracy, gap and relative gap (%) on Sudoku for Qwen-2 models. Only the 72B models can self-improve. For the 72B models, the improvement is around 200%.

|  | 0.5B | 1.5B | 7B | 72B |
|---|---|---|---|---|
| Generation Accuracy | 0.66 | 0.62 | 2.09 | 8.82 |
| gap | -0.09 | 0.04 | -0.07 | 16.99 |
| gap$_{rel}$ | -0.15 | -0.61 | -0.01 | 20.81 |

hold. One such scenario involves factual tasks that require generating a factually correct answer to a trivia question. We hypothesize that the capability to generate a correct answer is contingent solely on whether the model has been trained with the relevant factual knowledge, and verification would provide little additional signal. To test this, we measure $\mathsf{gap}(f)$ on the Natural Question dataset (Kwiatkowski et al., 2019), where $u(x, y) = 1$ if $y$ is one of the candidate answers to the question $x$, and $u(x, y) = 0$ otherwise. Our analysis on a test subset of 3610 questions, presented in Table 1, reveals that despite all models achieving non-trivial generation accuracy, the gap remains smaller than 1%, or is even negative, across all models. This suggests that certain tasks may not benefit from the current self-improvement framework. We include the full results in Table 5.

## 4.4 SUDOKU

Generalized sudoku is a canonical example where the generation (NP-hard) is harder than the verification (P) (Haythorpe, 2016). We consider 4 by 4 sudoku puzzles, each with a unique solution, with 288 puzzles in total. We task the models to use CoT reasoning for both generation and verification. The results, presented in Table 2 and detailed further in Appendix E.3, reveal a surprising pattern: only the largest models, such as Qwen-1.5/2 72B and Llama 3.1 70B, exhibit non-trivial gaps. For these models, the improvement is indeed more significant ($50\% - 300\%$ improvement in accuracy) than the improvement in the math task.

While the second observation aligns with common intuition, the first may be unexpected, as most models demonstrate the ability to self-improve on tasks where the gap between generation and verification appears even narrower, such as in GSM8K. We hypothesize that, despite sudoku verification being simpler than generation, it still necessitates a certain level of reasoning and planning, even with explicit verification guidelines. This requirement is similar in mathematical tasks; however, it is likely that most models have been exposed to math verification during pre-training, unlike sudoku verification. Consequently, smaller models may lack the requisite reasoning capabilities to improve on sudoku tasks. Although our analysis is primarily post-hoc, an interesting avenue for future research would be to develop a metric to predict a model's "self-improvability" on specific tasks.

---

**Takeaway on improvable tasks**

LLMs do not universally self-improve across all tasks:

- **Factual Tasks:** There is no significant generation-verification gap, given the similarity in complexity between generation and verification.

- **Sudoku:** Despite the exponential computational complexity separation between generation and verification in generalized sudoku, most models fail to self-improve. When improvement occurs, it is notably significant.

These findings suggest that the model's inherent reasoning and planning capabilities developed during pre-training are crucial for general self-improvement.

---

## 5 ITERATIVE SELF-IMPROVEMENT

Building on our understanding of single-round self-improvement, a natural extension is to study iterative self-improvement. As no additional information is introduced in the process, it is unrealistic to expect indefinite improvement. Thus in this section, we study the dynamics of the iterative self-improvement, and its relationship with model scales.

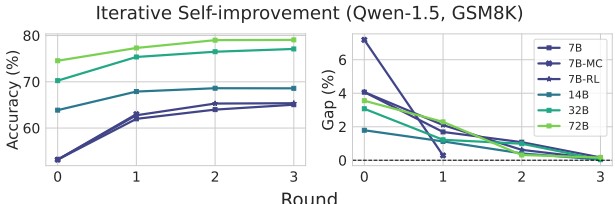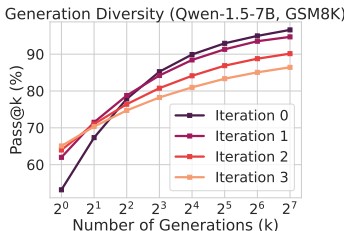

Figure 4: **Left**: The generation accuracy and gap along the iterative self-improvement process for Qwen-1.5 models with CoT-Binary and MC verification. RL denotes using RL for model update. The horizontal line on the gap plot denotes $0.5\%$. **Right**: The change of effective generation diversity along the iterative self-improvement process for Qwen-1.5 7B model, measured by pass@$k$ for different $k$.

In our experiment, we perform iterative self-improvement (Algorithm 1) on the Qwen-1.5 model family with CoT-Binary verification on GSM8K. We also perform model update using RL in addition to rejection sampling. We present the results in Figure 4 and defer the finetuning hyperparameters to Appendix F. We observe that 1) the gap diminishes nearly to zero within two or three rounds of self-improvement, consistent with the observation with previous works (Yuan et al., 2024; Liang et al., 2024). However, previous works failed to disentangle the gap and the model update. 2) The rate of saturation is similar across models with different capacities. 3) Notably, for the 7B and 14B models, the model accuracy at iteration one exceeds the sum of the generation accuracy and the gap at iteration 0, i.e., $J(f_1) > J(f_0[w(\widehat{u}_{f_0})])$. This increase is attributed to improved adherence to the required answer format post-finetuning – the discrepancy between "flexible match" and "exact match" (extract the answer from the required answer format) disappears after the first round. As we argued in the previous section, this additional accuracy gain is not due to the self-improvement capability of the model, and thus our modular study reduces the confounding factors in understanding the self-improvement capability of the model.

To compare the dynamics between verification methods, in Figure 4 we also plot MC (quantile 0.7) verification for the 7B model. We observe that the gap immediately drops to near 0 after the first round of self-improvement, and thus multi-round self-improvement with MC verification is unlikely. This rapid saturation is consistent across other thresholds for MC verification. We provide a more detailed study on the cause of this phenomenon in Section 6.1.

We also examine the "effective diversity" of generations throughout the iterative self-improvement process using the metric pass@$k$.[6] We present the results in Figure 4. We observe when $k$ is small, pass@$k$ increases with the number of rounds of self-improvement, validating the success of the self-improvement process. However, when $k$ is large, pass@$k$ decreases with the number of iterations, indicating that the diversity of the generations is reduced through the self-improvement process. This trend may result from the model's inability to verify rare, yet correct, answers, potentially leading to convergence on incorrect solutions during the self-improvement process.

For completeness, we also repeat the same experiments on the MATH benchmark. We defer the results to Appendix E.4. We observe the same phenomena of saturation limit and decrease in effective diversity in the MATH benchmark. In Appendix D.3, we also perform experiments on iterative improvement with a more powerful verifier and ground-truth labels.

> **Takeaway on iterative self-improvement**
>
> LLMs can perform iterative self-improvement with an effective verification method:
>
> - **Saturation Limit:** Without new information, iterative self-improvement typically saturates after two or three rounds, regardless of the model's capacity.
> - **Cause of Saturation:** A potential reason for this saturation is a decrease in effective diversity, caused by convergence on incorrect answers for certain questions.
>
> Addressing the reduction in diversity could potentially extend the duration and effectiveness of the self-improvement process.

---

[6]Given a question, pass@$k$ is 1 if at least one of the $k$ generations of the model is correct, or 0 otherwise.

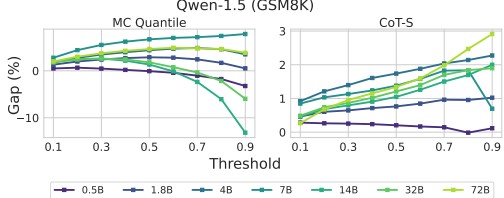
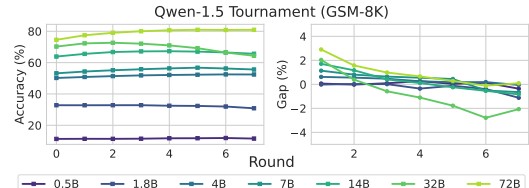

Figure 5: Change in the gap as the threshold varies for each verification method. We only present the results for MC with quantile threshold and CoT-Score, because CoT-Binary's gap stays constant as the threshold varies.

Figure 6: Change in the generation accuracy of the filtered dataset (left) and gap (right) with respect to the round of tournament. The right figure plots the gap with respect to the accuracy from the previous round instead of the base accuracy.

## 6 A FINE-GRAINED STUDY ON VERIFICATION

Among the three components of self-improvement, the verification step offers the most flexibility, whereas generation and update follow more fixed procedures. Therefore, this section presents a detailed examination of the verification mechanisms. Through this particular study, we aim to uncover practical ways to enhance the overall self-improvement process.

### 6.1 GENERALIZATION OF VERIFICATIONS

In the rejection sampling framework, selecting an appropriate threshold for filtering generations based on verification is a crucial practical concern. We explore verification methods adaptable to various thresholds, including MC, CoT-Score, and Tournament. Our analysis focuses on how the gap changes with different thresholds for these methods. We present results for the Qwen-1.5 models in Figs. 5 and 6. For tournament verification, the threshold is defined as the number of rounds of tournament. We defer the results of other models to Appendix E.5. In Tournament, we note that the gap with respect to the accuracy in the previous iteration generally decreases monotonically; this trend occurs as the verification error is more likely to be exploited by the remaining generations at later stages of the tournament.

We also conduct two additional sanity checks: in Figure 16, we plot the distribution of scores with CoT-Score verification, and the mode of the score is at 10 for all models. This self-bias behavior is expected and it is also observed in Xu et al. (2024). We also check the recency bias (Zhao et al., 2021) in Tournament verification: the average probability of preferring the first generation across all models and all rounds is $0.56$ with a standard deviation of $0.12$, indicating no critical recency bias.

We observe that the relationship between the gap and the threshold is consistent across most models when using any fixed verification method. For MC and Tournament, the gap follows a concave curve relative to the threshold, while for CoT-Score, it increases monotonically. In addition, most models agree on the optimal thresholds for each verification across model families, and we use the optimal thresholds to report our results for this paper. However, in general, one should not expect the optimal threshold transfers between different tasks. That said, the consistency in threshold effects suggests a practical approach: if determining the optimal threshold for a large model is costly, one might first establish it for a smaller model and then apply it to the larger model.

### 6.2 CORRELATION STUDIES BETWEEN VERIFICATION METHODS

As the verification methods are functionally similar, one might question the necessity to study multiple verifications. To address this, We start by comparing the distribution of gaps induced by different verification methods. We present the bar plot of the gap distribution of Qwen-1.5 7B in Figure 7. Notably, there are significant discrepancies, especially between MC and CoT methods – the variance in the gap is considerably larger with MC. This aligns with our previous findings in Section 5 where iterative self-improvement with MC verification saturates more quickly than with CoT. While CoT slightly improves the accuracy on most questions, MC will drive the accuracy to the extreme in one round of self-improvement. We observe that this pattern holds across all models, as detailed in Figure 17.

Figure 7: **Left:** The empirical distribution of gaps of MC Quantile and CoT-Binary of Qwen-1.5-7B on GSM8K. We cluster gaps in bins of intervals with width of 0.005. We label the mean ($\mu$) and standard deviation ($\sigma$) of each distribution. **Right:** the correlation plot of the output of each verification $\hat{u}$ and the correlation plot of the gap from each verification and generation accuracy.

To further compare the verification methods, we calculate the Pearson correlation coefficient between the outputs of the proxy utility $\hat{u}$ the gaps, shown in Figure 7. We use Qwen-1.5 7B as an example and defer full results to Figs. 18 and 19. We observe that the correlations between $\hat{u}$ are generally low, suggesting potential benefits in combining different verification methods. Notably, the correlation between MC verifications and the correlation between CoT verifications are generally the highest, and larger models tend to have a higher correlation between the gaps. Surprisingly, the gaps of any verification method do not positively correlate with generation accuracy, reinforcing the idea that the relative gap may be a more appropriate metric for measuring self-improvement capability.

### 6.3 IMPROVEMENT VIA ENSEMBLE

The non-overlap property of different verification methods suggests the potential for enhanced verification performance through their combination. We again focus on the rejection sampling setup. We employ a logical AND operation, keeping samples only if they pass all verification filters. The results are deferred to Appendix E.7. We observe that combining any verifications with non-trivial gaps improves the verification performance (with the exception of CoT for 0.5B model with near 0 gap). This promising outcome indicates that despite functional similarities, different verification mechanisms can still be combined to increase self-improvement efficacy. The consistent improvements across different model sizes also suggest that strategies developed using smaller models can be effectively applied to larger ones, if all verifications are valid.

> **Takeaway on verification mechanisms**
>
> A fine-grained study on verification reveals several implications for practice:
>
> - **Verification Consistency:** The distribution of the gaps and optimal verification threshold typically generalize across models.
> - **Verification Distinction:** Despite functional similarities, the outputs and gaps of verification methods show non-trivial differences among each other.
> - **Ensemble Heuristic:** Simple verification ensemble heuristics can improve performance.
>
> The consistency result suggests that configurations from smaller models can be applied to larger ones to avoid the costs associated with tuning big models. The discovery that simple ensemble techniques can enhance performance highlights the potential for more sophisticated algorithms to advance self-improvement strategies further.

### 7 CONCLUSION AND DISCUSSION

In this paper, we conduct comprehensive and controlled studies on the LLM self-improvement framework through multiple model families, tasks, and verification mechanisms. We structure the mathematical framework of the self-improvement process and pinpoint the generation-verification gap as a critical metric. Our results reveal several intriguing properties such as the scaling properties of the relative gap, saturation of iterative self-improvement and enhancement of verification via ensemble methods. These insights are likely to have practical implications for improving pre-training, post-training, and test-time inference. Additionally, our research opens several promising avenues for future exploration, and we defer the list to Appendix A.

ACKNOWLEDGEMENT

The authors are grateful to Audrey Huang and Akshay Krishnamurthy for constructive discussion. The authors thank Cyril Zhang for helpful feedback on the draft. The authors thank Riccardo Savorgnan, Sohrab Andaz and other members of the Amazon SCOT-RL team for discussion and infrastructure support. HZ is supported by an Eric and Susan Dunn Graduate Fellowship. SK acknowledges the support of the Chan Zuckerberg Initiative Foundation for establishing the Kempner Institute for the Study of Natural and Artificial Intelligence, the Office of Naval Research under award N00014-22-1-2377, and the National Science Foundation under award #IIS 2229881.

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

# A FUTURE DIRECTIONS

In this section we list several promising avenues for future exploration:

- While our scaling analysis is primarily observational (Ruan et al., 2024), pursuing a more extensive scaling law study (Kaplan et al., 2020) based on our preliminary findings could provide robust empirical guidelines.

- Our results hint at an inference-time scaling law (Wu et al., 2024) is possible for self-improvement (or with cross-improvement (c.r. Section 4.2)). Identifying compute-optimal methods for self-improvement across different tasks remains a critical challenge.

- The decline in the effective diversity of generations during iterative self-improvement presents a significant obstacle. Developing strategies to mitigate this issue offers considerable empirical benefits.

- The distinct non-overlap property of verification mechanisms, despite their functional similarities, suggests that combining compositional verification could significantly enhance self-improvement. Exploring this potential further could yield fruitful results.

# B RELATED WORK

**Synthetic Data and Self-Training.** Training LLMs with a mixture of "real" data (generated by human) and synthetic data has been the standard protocol nowadays given the limited number of human data and extensive amount of data required as we scale up the LLM training. Initial studies generated synthetic data from more powerful models (Gunasekar et al., 2023; Li et al., 2023; Team et al., 2023; Sun et al., 2023; Taori et al., 2023; Zhu et al., 2023; Wei et al., 2024), while recent approaches involve models training on their own outputs (Achiam et al., 2023; Adler et al., 2024; Dubey et al., 2024; Yang et al., 2024; Hui et al., 2024).

On the theoretical front, extensive research has explored the phenomenon of model collapse during self-training and strategies to counter this degenerate behavior (Hataya et al., 2023; Martínez et al., 2023; Bertrand et al., 2023; Briesch et al., 2023; Taori & Hashimoto, 2023; Alemohammad et al., 2023; Dohmatob et al., 2024; Gillman et al., 2024).

**LLM Self-improvement.** One of the most effective strategies to prevent model collapse during self-training is the use of a reliable verifier (Gillman et al., 2024). In the absence of additional resources like labeled data or an external oracle, models can utilize their own verification capabilities. This is particularly effective if the model is more proficient at verification than generation. Numerous studies have proposed variations of self-improvement algorithms based on this principle, resulting in significant practical achievements (Zelikman et al., 2022; Wang et al., 2022b; Huang et al., 2022; Singh et al., 2023; Chen et al., 2023; Madaan et al., 2024; Xu et al., 2024; Yuan et al., 2024; Liang et al., 2024; Wang et al., 2024b; Shinn et al., 2024; Zelikman et al., 2024; Chen et al., 2024; Jiang et al., 2024). Previous research, however, often relied on additional data to enhance verification, used surrogate metrics for improvement, or limited their focus to a small number of models. In this work, instead of proposing any new algorithm, we aim to rigorously analyze the self-improvement phenomenon in a controlled, comprehensive manner. On the theoretical front, Huang et al. (2025) studied self-improvement using trajectory probability as score, and proved the optimality of SFT (rejection sampling) with certain coverage conditions.

**Improving Test-time Inference with Additional Computation.** Recent research has demonstrated that the performance of models can be enhanced by allocating more computational resources to inference (Welleck et al., 2024; Damani et al., 2024). This typically leverages the observation that LLMs can make diverse generations, and with a small probability it can generate high-quality responses (Li et al., 2022; Brown et al., 2024; Bansal et al., 2024). Thus with oracle verifier, or with training a high-quality reward model, model performance can be improved by simply making multiple generations and selecting the best ones according to the oracle or the reward model (Cobbe et al., 2021). There are also works on training process-based reward models (Lightman et al., 2023) to improve the model's reasoning results (Luo et al., 2024; Wang et al., 2024a; Zhang et al., 2024a).

Concurrently there are also works on the test-time scaling law which investigates the computational trade-off between the model size (which determines the number of generations given a computation budget) and final accuracy combined with reward model or oracle (Wu et al., 2024; Snell et al., 2024).

The results provide the compute-optimal solution for test-time inferencing with a fixed compute budget and a fixed verifier. We believe a better understanding of self-improvement can also lead to a test-time scaling law without an external verifier.

**LLM-as-a-Judge.** LLM-as-a-judge refers to using an LLM to verify the generation of some other (or the same) LLM (Chiang et al., 2023; Zheng et al., 2023; Bubeck et al., 2023; Chiang & Lee, 2023; Zhou et al., 2024). Recently the same idea has also been applied to train a generative reward model (Ankner et al., 2024; Zhang et al., 2024b). Having a model that can verify its own generation is one of the key components of self-improvement, and in this work, we perform a fine-grained study on various types of LLM verification mechanisms.

**Reranking Algorithms.** The self-improvement framework we study in this paper relies on reweighting the generation distribution. Prior to self-improvement, the reranking algorithm has already been widely applied in various NLP applications (Collins & Koo, 2005; Huang & Chiang, 2007; Stiennon et al., 2020; Cobbe et al., 2021; Krishna et al., 2022; Lightman et al., 2023).

## C   Verification Mechanisms

In this section, we provide a more complete description of the verification mechanism we use throughout the paper.

- **Multiple Choice (MC)**: Multiple choice verification asks the LM to label responses as "Correct" and "Incorrect". Let $\mathsf{prom}_{\mathsf{mc}}$ be a verification prompt and denote $\widehat{u}_f^{\mathsf{mc}}(x, y)$ a utility derived from the verifier generating a single token $t^+, t^-$, representing the word "Correct" and "Incorrect" respectively. The score uses the logits from these tokens to find the probability of "Correct" conditioned on the next token being "Correct" or "Incorrect":

$$\widehat{u}_f^{\mathsf{mc}}(x, y) := \frac{f(t^+ \mid x, y, \mathsf{prom}_{\mathsf{mc}})}{f(t^+ \mid x, y, \mathsf{prom}_{\mathsf{mc}}) + f(t^- \mid x, y, \mathsf{prom}_{\mathsf{mc}})} \ .$$

- **Chain of Thought (CoT)**: CoT verification asks the LM to score responses and to provide the CoT. Denote by $\mathcal{S} \subset \mathbb{R}$ the set of verification scores and by $\mathsf{prom}_{\mathcal{S}}$ a verification prompt. We can define a utility

$$\widehat{u}_f^{\mathcal{S}}(x, y) := \mathbb{E}_{s, z \sim f(\cdot \mid x, y, \mathsf{prom}_{\mathcal{S}})}[s(z)],$$

where $z$ is the verification CoT, and $s(z) \in \mathcal{S}$ is the score extracted from the CoT. In our experiments we consider two versions, CoT-Score with $\mathcal{S} = [10]$ and CoT-Binary with $\mathcal{S} = \{0, 1\}$.

- **Tournament (To)**: The tournament verification does not directly fit the utility framework described in Section 2. Rather, this verification procedure involves comparisons of a batch of generations to provide a modified distribution[7]. Given a comparison prompt $\mathsf{prom}_{\mathsf{com}}$, we perform a tournament-style elimination over a batch of $2^r$ generations by comparing disjoint pairs in each round until a single generation remains. Let $\mathcal{Y}^{(0)} = y_1, y_2, \ldots, y_{2^r}$ be the initial set of generations. At round $k$, the set $\mathcal{Y}^{(k)}$ contains $2^{r-k}$ remaining generations. These are split into disjoint pairs $(y_i, y_j) \in \mathcal{Y}^{(k)}$. Each pair is compared using the prompt $\mathsf{prom}_{\mathsf{com}}$, and the verifier's output $s \in A, B$ indicates the preferred generation:

$$y_{\mathsf{win}} = \begin{cases} y_i & \text{if } f(\cdot \mid y_i, y_j, \mathsf{prom}_{\mathsf{com}}) = A, \\ y_j & \text{if } f(\cdot \mid y_i, y_j, \mathsf{prom}_{\mathsf{com}}) = B. \end{cases}$$

where $y_{\mathsf{win}}$ is the winner of the pairwise comparison and advances to the next round. After each round $k$, the set of winners $\mathcal{Y}^{(k+1)}$ contains half the number of generations. This process is repeated until $k = r$, leaving only one generation, the lone element in $\mathcal{Y}^r$.

---

[7]This batch-style distribution weighting also applies to strategies like top-k wherein we take the highest $k$ utility generations for a particular question.

# D ABLATION RESULTS

## D.1 SAMPLING TEMPERATURE

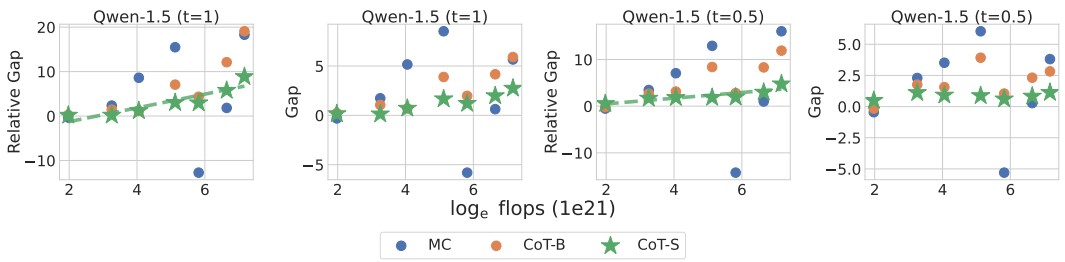

Figure 8: With proper verification method (e.g., CoT-S), with two different temperatures within a reasonable range, *the relative generation-verification gap (Definition 2.2) still scales monotonically with respect to the pre-training flops.* We conjecture that in this case, the relative gap is linear with respect to the log of the pre-training flops. Again, we do not observe scaling phenomenon for generation-verification gap.

In this section, we tested the robustness of our result with different sampling parameters. Specifically, we tried temperature $t = 1$ and $t = 0.5$ for both generation and verification (the default temperature is $0.7$). We repeat the scaling experiment on GSM8K with Qwen-1.5 model family (which contains the most number of models), and we record the scaling of gap and relative gap in Figure 8. We observe the same scaling phenomena in both temperatures that the relative gap scales monotonically with the pretrain flops and is almost linear with respect to the log of the pretrain flops. The only exception is the 14B model with temperature 1, but we believe this is due to higher temperature introducing more noise and more samples of generation or verification will recover the perfect trend. We also observe that the improvement in temperature 0.5 setting is less prominent, and we believe this is due to the reduced diversity in the generation with a lower temperature.

## D.2 INSTRUCT MODELS

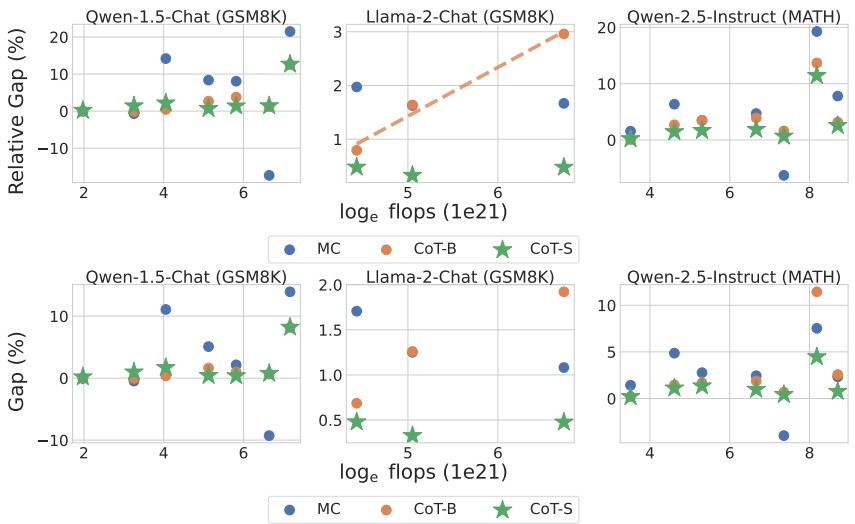

Figure 9: Instruct models do not always have the scaling property.

In this section, we present our results on examining the scaling property of the instruct models. Specifically, we tested Qwen-1.5-Chat and Llama-2-Chat model family on GSM8K and Qwen-2.5-Instruct model family on MATH. We observe that Qwen-1.5-Chat models do not have the scaling property. In fact, even the accuracy of the models do not scale monotonically with respect to the

model size (and our accuracy results match the reported number in Bai et al. (2023)). Similarly, we does not observe the scaling property on Qwen-2.5-Instruct model family as well. On the other hand, Llama-2-Chat models demonstrate the scaling property but we remark that the curve fit is only based on three models.

In some sense, this result is expected. As we discussed in the earlier sections, instruct models have more confounders that may not lead to a clear conclusion. In addition, some latest models such as Llama 3.1 already have a self-improvement component in the finetune process so it is unclear if additional self-improvement signals will be observed. This result justifies our choice of using base models to study self-improvement.

## D.3 FIXED VERIFIER

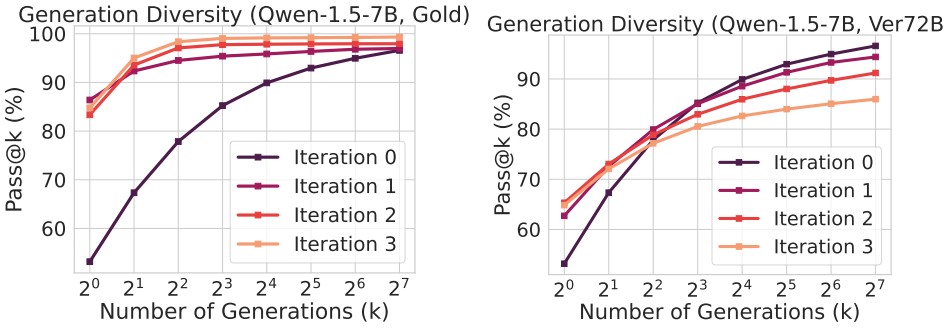

Figure 10: The change of effective generation diversity along the iterative self-improvement process for Qwen-1.5 7B model, measured by pass@$k$ for different $k$. **Left**: using gold verifier. **Right**: using Qwen-1.5-72B as the verifier.

To achieve a better understanding for the reason that effective diversity degrades during self-improvement, in this section we conduct an ablation experiment where we fix the verifier unchanged. We consider two settings, the first is to use the gold (ground truth) verifier/label and the second is to use a more powerful model. Following previous protocols, we use Qwen-1.5-7B model on GSM8K. We record the results in Figure 10.

We observe that, with gold labels, there is no degradation of the effective diversity along the iterative training. This is because with the gold label, all the incorrect generations will be filtered out and thus the model will not concentrate on any wrong answers, which is the most intuitive cause of the degradation of the effective diversity. Meanwhile, the same degradation in the effective diversity still happens if we use a more powerful but imperfect verifier because in some questions, the model will concentrate on some wrong answers which the verifier labels correct.

## D.4 VARIANCE OF THE VERIFIER

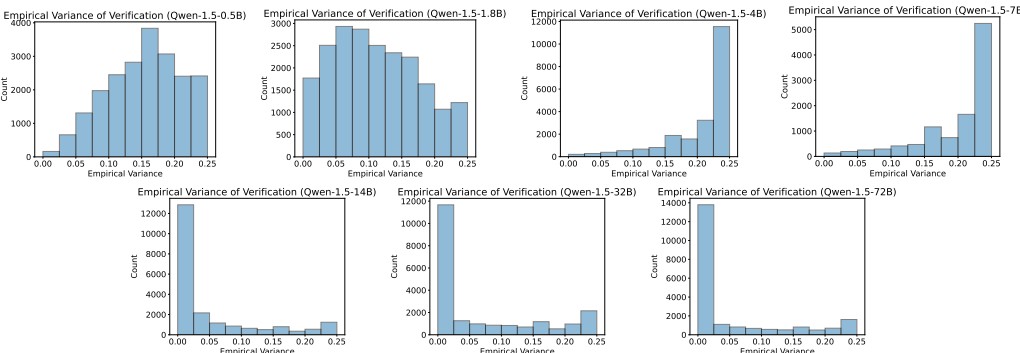

Figure 11: Histograms of variance of generation from Qwen-1.5 model family in GSM8K.

In this section, we examine the effect of a single generation sample on the evaluation of generation-verification gap. The metric we use is the variance of the generation. Specially, we tested Qwen-1.5-72B model on GSM8K, where we use the same sampling hyperparameter ($t = 0.7, p = 0.9$) as the main text. For each question, we generate 8 responses, and for each response, we generate 64 verifications with CoT-B generation. Fix each generation, we can treat the distribution of the CoT-B generation as a Bernoulli distribution with parameter $p$, where $p$ is the probability of the verification ending with "correct". For each generation, we measure the empirical variance ($p(1 - p)$) and present the histogram in Figure 11. Note that the variance ranges from 0 ($p = 1$ or $p = 0$) to 0.25 ($p = 0.5$). We observe that the variance behaves differently for the models, but for bigger models, for most generations, the verification variance is near 0.

# E    ADDITIONAL RESULTS

## E.1    ADDITIONAL RESULTS FOR SECTION 4.1

Table 3: Gap on GSM-8K for all models. For each verification, "top $n$" denotes taking the threshold as the $n$ quantile of the proxy utility for each prompt, and $\tau = n$ denotes taking the threshold as $n$ for all prompts. All numbers denote the percentage.

| Name | Size | Accuracy | MC | | | | CoT | | | To |
| | | | top 0.7 | top 0.8 | $\tau = 0.7$ | $\tau = 0.8$ | Bin | S ($\tau = 8$) | S ($\tau = 9$) | round 5 |
|---|---|---|---|---|---|---|---|---|---|---|
| Qwen-1.5 | 0.5B | 11.31 | -0.62 | -1.42 | -2.27 | -3.68 | -0.02 | -0.01 | 0.12 | 0.44 |
| | 1.8B | 32.81 | 3.06 | 2.70 | -0.01 | 0.00 | 1.58 | 0.95 | 1.02 | -0.44 |
| | 4B | 50.21 | 4.85 | 4.63 | 0.48 | 0.89 | 2.36 | 2.14 | 2.27 | 2.08 |
| | 7B | 53.17 | 7.18 | 7.42 | 3.68 | 6.13 | 4.07 | 1.84 | 0.70 | 3.59 |
| | 14B | 63.87 | -2.04 | -5.61 | 2.36 | -1.06 | 1.79 | 1.69 | 2.00 | 3.16 |
| | 32B | 70.25 | -0.22 | -1.98 | 1.72 | 1.06 | 3.07 | 1.84 | 1.90 | -1.03 |
| | 72B | 74.55 | 4.84 | 4.75 | 2.00 | 3.95 | 3.55 | 2.47 | 2.91 | 6.40 |
| Qwen-2 | 0.5B | 26.19 | 4.59 | 3.39 | 3.81 | -6.59 | 0.21 | 0.64 | 0.72 | 0.32 |
| | 1.5B | 48.82 | 6.09 | 5.27 | 5.02 | 1.32 | 1.26 | 2.78 | 2.80 | 1.29 |
| | 7B | 76.42 | 4.69 | 4.66 | 3.90 | 2.17 | 2.36 | 1.97 | 2.08 | 3.86 |
| | 72B | 81.69 | 2.39 | 2.38 | 0.89 | 2.12 | 2.51 | 2.08 | 2.45 | 2.20 |
| Llama-2 | 7B | 11.64 | 2.33 | 2.20 | 0.10 | 0.00 | 0.16 | 0.25 | 0.23 | 0.99 |
| | 13B | 21.57 | 3.45 | 3.18 | -0.13 | 0.01 | 1.13 | 0.97 | 1.01 | 1.17 |
| | 70B | 48.42 | 5.14 | 4.91 | 4.77 | 4.16 | 3.98 | 3.44 | 3.45 | 5.68 |
| Llama-3 | 8B | 45.66 | 5.34 | 5.33 | -0.50 | 0.44 | 2.67 | 2.10 | 2.13 | 4.08 |
| | 70B | 74.19 | 5.06 | 4.52 | 4.68 | 1.35 | 3.89 | 2.59 | 2.72 | 2.00 |
| Llama-3.1 | 8B | 49.31 | 4.68 | 4.57 | -2.25 | -0.24 | 3.37 | 2.09 | 2.05 | 4.78 |
| | 70B | 71.71 | 6.88 | 6.77 | 6.00 | 0.93 | 3.29 | 2.71 | 2.88 | -0.40 |
| Yi-1.5 | 6B | 55.53 | 4.40 | 4.27 | 2.98 | -0.97 | 2.01 | 1.88 | 1.95 | 2.24 |
| | 9B | 61.04 | 7.74 | 7.50 | 5.61 | 5.73 | 2.32 | 3.61 | 3.72 | 7.83 |
| | 34B | 73.71 | 6.29 | 6.23 | 3.34 | 4.84 | 2.49 | 2.86 | 2.95 | 3.41 |

Table 4: Relative gaps on GSM-8K for all models. For each verification, "top $n$" denotes taking the threshold as the $n$ quantile of the proxy utility for each prompt, and $\tau = n$ denotes taking the threshold as $n$ for all prompts. All numbers denote the percentage.

| Name | Size | Accuracy | MC | | | | CoT | | | To |
| | | | top 0.7 | top 0.8 | $\tau = 0.7$ | $\tau = 0.8$ | Bin | S ($\tau = 8$) | S ($\tau = 9$) | round 5 |
|---|---|---|---|---|---|---|---|---|---|---|
| Qwen-1.5 | 0.5B | 11.31 | -0.70 | -1.60 | -2.56 | -4.15 | -0.03 | -0.13 | 0.04 | 0.59 |
| | 1.8B | 32.81 | 4.55 | 4.01 | -0.01 | -0.01 | 4.45 | 3.10 | 3.15 | -0.39 |
| | 4B | 50.21 | 9.75 | 9.31 | 0.96 | 1.80 | 8.22 | 6.92 | 8.12 | 5.24 |
| | 7B | 53.17 | 15.34 | 15.84 | 7.87 | 13.08 | 14.06 | 5.50 | -0.35 | 10.26 |
| | 14B | 63.87 | -5.64 | -15.54 | 6.52 | -2.94 | 7.77 | 8.00 | 9.02 | 12.19 |
| | 32B | 70.25 | -0.75 | -6.67 | 5.78 | 3.57 | 15.31 | 8.40 | 8.49 | -4.55 |
| | 72B | 74.55 | 19.01 | 18.65 | 7.87 | 15.52 | 21.82 | 14.96 | 16.97 | 32.06 |
| Qwen-2 | 0.5B | 26.19 | 6.22 | 4.59 | 5.16 | -8.93 | 0.44 | 1.58 | 1.79 | 1.31 |
| | 1.5B | 48.82 | 11.90 | 10.30 | 9.82 | 2.58 | 4.45 | 9.96 | 10.11 | 5.26 |
| | 7B | 76.42 | 19.89 | 19.74 | 16.53 | 9.18 | 16.46 | 13.73 | 14.34 | 17.17 |
| | 72B | 81.69 | 13.07 | 13.00 | 4.87 | 11.57 | 20.16 | 14.34 | 19.92 | 17.84 |
| Llama-2 | 7B | 11.64 | 2.64 | 2.49 | 0.11 | 0.00 | 0.25 | 0.39 | 0.37 | 1.91 |
| | 13B | 21.57 | 4.40 | 4.05 | -0.16 | 0.02 | 2.45 | 1.79 | 1.82 | 2.88 |
| | 70B | 48.42 | 9.97 | 9.52 | 9.25 | 8.07 | 13.77 | 12.22 | 12.07 | 18.57 |
| Llama-3 | 8B | 45.66 | 9.62 | 9.60 | -0.90 | 0.80 | 8.51 | 6.16 | 6.70 | 12.55 |
| | 70B | 74.19 | 18.08 | 16.13 | 16.71 | 4.84 | 19.09 | 13.24 | 13.74 | 11.28 |
| Llama-3.1 | 8B | 49.31 | 8.97 | 8.77 | -4.31 | -0.47 | 11.74 | 7.02 | 6.89 | 14.97 |
| | 70B | 71.71 | 22.83 | 22.47 | 19.91 | 3.09 | 16.68 | 12.65 | 13.59 | -1.03 |
| Yi-1.5 | 6B | 55.53 | 9.89 | 9.60 | 6.70 | -2.17 | 9.10 | 6.60 | 6.69 | 10.64 |
| | 9B | 61.04 | 19.86 | 19.25 | 14.40 | 14.70 | 10.09 | 14.29 | 14.35 | 24.78 |
| | 34B | 73.71 | 23.94 | 23.68 | 12.70 | 18.40 | 15.02 | 16.69 | 16.96 | 8.89 |

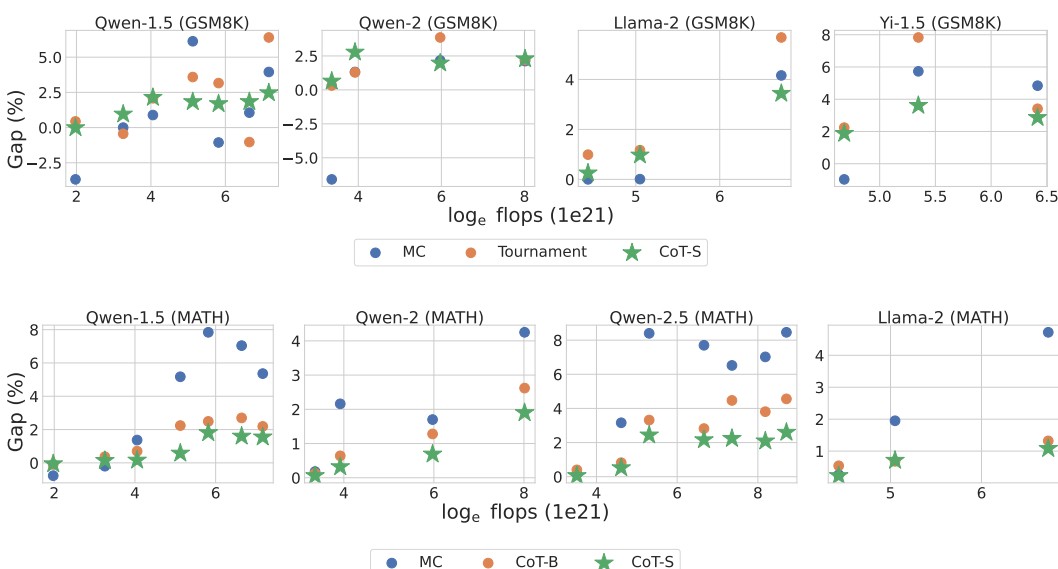

Figure 12: No clear scaling phenomenon of the gap with respect to the pretrain flops. The x-axis is the log of pretrain flops, and the y-axis is the relative gap. MC denotes Multiple Choice verification with quantile threshold 0.8 (top 0.8), CoT-S denotes CoT-Score verification with global threshold 8, and To denotes Tournament verification with 5 rounds.

## E.2 ADDITIONAL RESULTS FOR SECTION 4.3

| Model | Qwen-1.5 | | | | | | |
|---|---|---|---|---|---|---|---|
| | 0.5B | 1.8B | 4B | 7B | 14B | 32B | 72B |
| Generation Accuracy | 6.20 | 11.40 | 17.16 | 21.20 | 26.83 | 35.79 | 39.97 |
| MC (top 0.2) | -0.62 | -0.02 | -1.20 | -1.15 | -1.99 | -0.43 | -0.75 |
| MC ($\tau = 0.8$) | -0.51 | -0.32 | -0.72 | -1.07 | -1.21 | -0.10 | 0.32 |

| Model | Qwen-2 | | | |
|---|---|---|---|---|
| | 0.5B | 1.5B | 7B | 72B |
| Generation Accuracy | 6.51 | 13.87 | 29.09 | 41.45 |
| MC (top 0.2) | -0.06 | 0.04 | 0.79 | 0.28 |
| MC ($\tau = 0.8$) | -0.05 | 0.02 | -0.05 | -0.05 |

| Model | Llama-2 | | |
|---|---|---|---|
| | 7B | 13B | 70B |
| Generation Accuracy | 25.52 | 41.00 | 29.09 |
| MC (top 0.2) | -0.96 | 0.76 | 0.30 |
| MC ($\tau = 0.8$) | -0.81 | -2.31 | -0.44 |

| Model | Llama-3 | |
|---|---|---|
| | 8B | 70B |
| Generation Accuracy | 30.40 | 45.59 |
| MC (top 0.2) | 0.27 | 0.32 |
| MC ($\tau = 0.8$) | -0.23 | -0.41 |

| Model | Llama-3.1 | |
|---|---|---|
| | 8B | 70B |
| Generation Accuracy | 27.75 | 45.13 |
| MC (top 0.2) | 0.42 | 0.44 |
| MC ($\tau = 0.8$) | -0.59 | -0.37 |

| Model | Yi-1.5 | | |
|---|---|---|---|
| | 6B | 9B | 34B |
| Generation Accuracy | 22.82 | 25.94 | 35.31 |
| MC (top 0.2) | 0.09 | 0.21 | 0.24 |
| MC ($\tau = 0.8$) | -0.07 | 0.61 | 0.30 |

Table 5: Gap on Natural Question for all models. With non-trivial generation accuracy, all gaps are near 0, indicating that the task is non-improvable.

## E.3 ADDITIONAL RESULTS FOR SECTION 4.4

| Model | Qwen-1.5 | | | | | | |
|---|---|---|---|---|---|---|---|
| | 0.5B | 1.8B | 4B | 7B | 14B | 32B | 72B |
| Generation Accuracy | 0.43 | 1.00 | 0.88 | 0.95 | 1.57 | 2.67 | 2.02 |
| Gap | 0.02 | -0.03 | -0.15 | -0.64 | 0.22 | 0.07 | 1.23 |
| Relative Gap | -0.10 | -2.80 | -1.39 | -3.06 | 0.67 | -1.25 | 1.14 |

| Model | Qwen-2 | | | | | |
|---|---|---|---|---|---|---|
| | 0.5B | 1.5B | 7B | 72B | 7B-Instruct | 72B-Instruct |
| Generation Accuracy | 0.66 | 0.62 | 2.09 | 8.82 | 2.16 | 8.15 |
| Gap | -0.09 | 0.04 | -0.07 | 16.99 | 0.13 | 22.97 |
| Relative Gap | -0.15 | -0.61 | -0.01 | 20.81 | 0.20 | 26.40 |

| Model | Llama-2 | | |
|---|---|---|---|
| | 7B | 13B | 70B |
| Generation Accuracy | 0.82 | 0.89 | 0.86 |
| Gap | -0.13 | -0.63 | -0.86 |
| Relative Gap | 0.45 | -2.02 | -3.57 |

| Model | Llama-3 | |
|---|---|---|
| | 8B | 70B |
| Generation Accuracy | 1.39 | 1.63 |
| Gap | -1.10 | -0.84 |
| Relative Gap | -15.4 | -36.12 |

| Model | Llama-3.1 | |
|---|---|---|
| | 8B | 70B |
| Generation Accuracy | 1.11 | 1.68 |
| Gap | -0.19 | 5.5 |
| Relative Gap | -4.52 | 6.87 |

| Model | Yi-1.5 | | |
|---|---|---|---|
| | 6B | 9B | 34B |
| Generation Accuracy | 0.59 | 1.29 | 4.48 |
| Gap | -0.60 | 0.22 | -1.75 |
| Relative Gap | -0.94 | 0.43 | -0.77 |

Table 6: Generation accuracy, gap and relative gap on Sudoku for all models.

### E.4 ADDITIONAL RESULTS FOR SECTION 5

In this section we repeat the experiments in Section 5 on the MATH dataset. We observe similar results as in Section 5: iterative self-improvement saturates (in 4 iterations) with gap diminishing to 0. We record a partial result at the top of Figure 13. We also observe the same degradation in effective diversity along the iterative self-improvement and we record the result in the bottom of Figure 13.

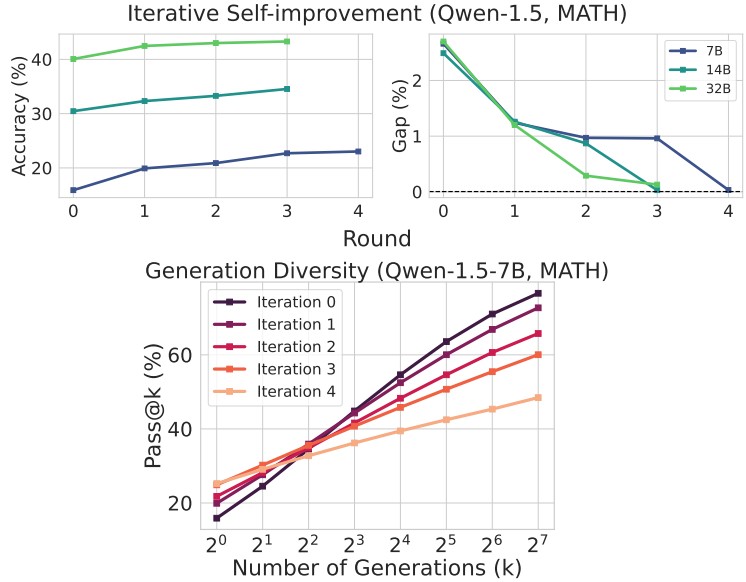

Figure 13: **Top**: The generation accuracy and gap along the iterative self-improvement process for Qwen-1.5 model family with CoT-Binary. The horizontal line on the gap plot denotes $0.5\%$. **Bottom**: The change of effective generation diversity along the iterative self-improvement process for Qwen-1.5 7B model, measured by pass@$k$ for different $k$.

### E.5    ADDITIONAL RESULTS FOR SECTION 6.1

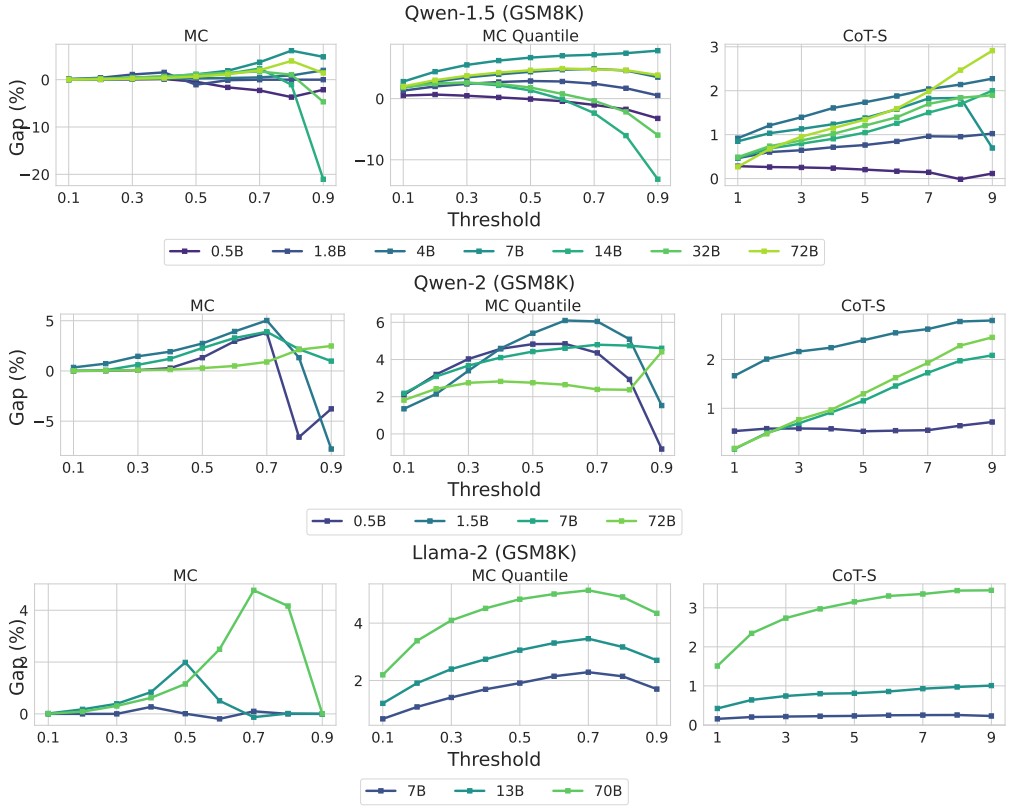

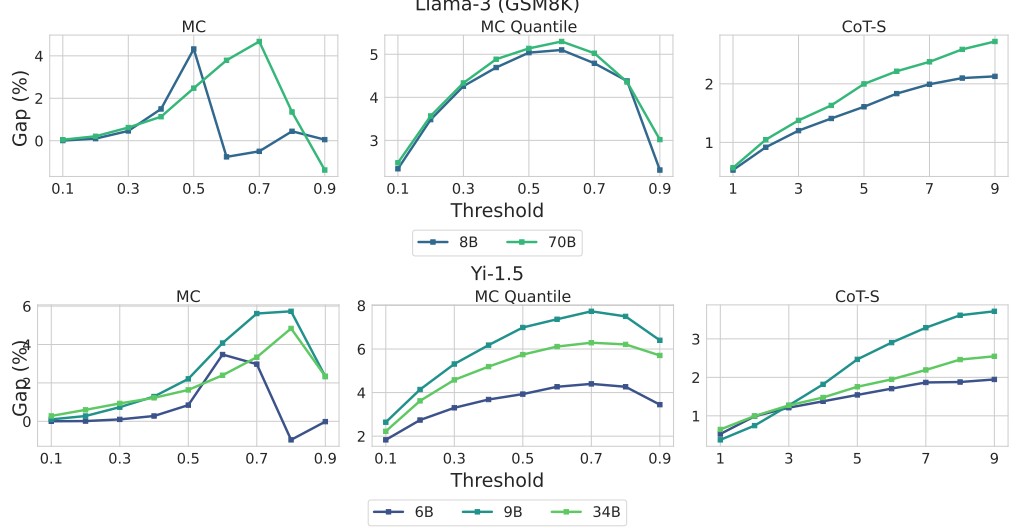

Figure 14: Change in the gap as we vary the threshold for each verification method. We only present the results for MC with global threshold, quantile threshold and CoT-Score, because CoT-Binary's gap does not change as we change the threshold.

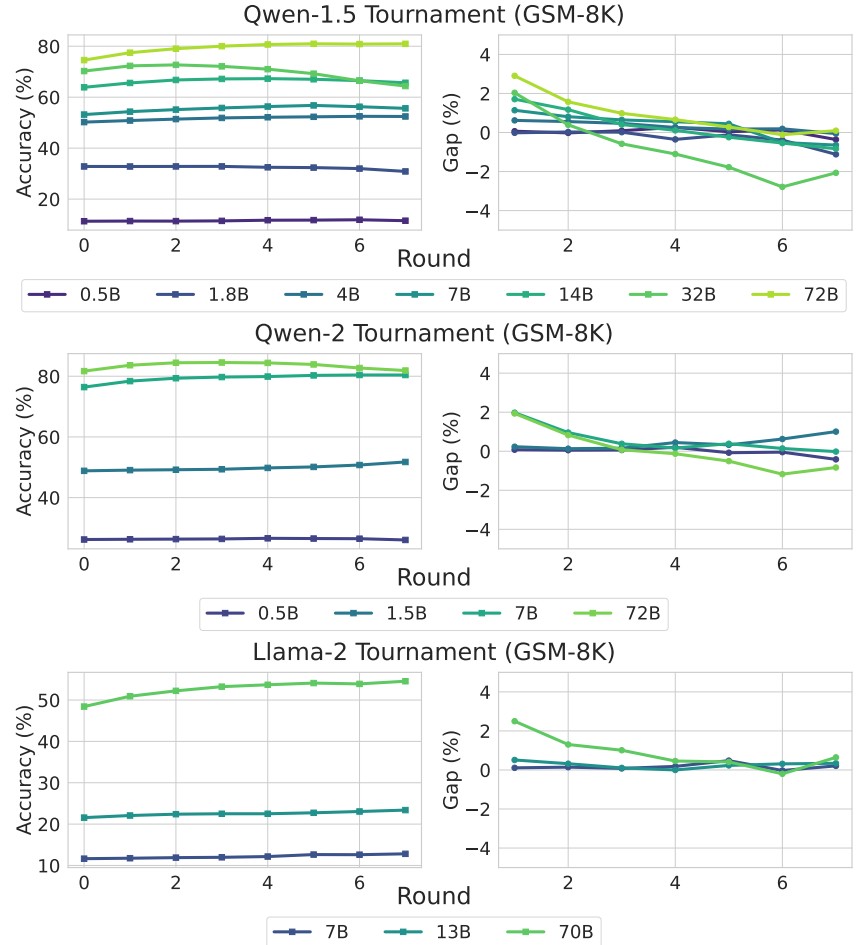

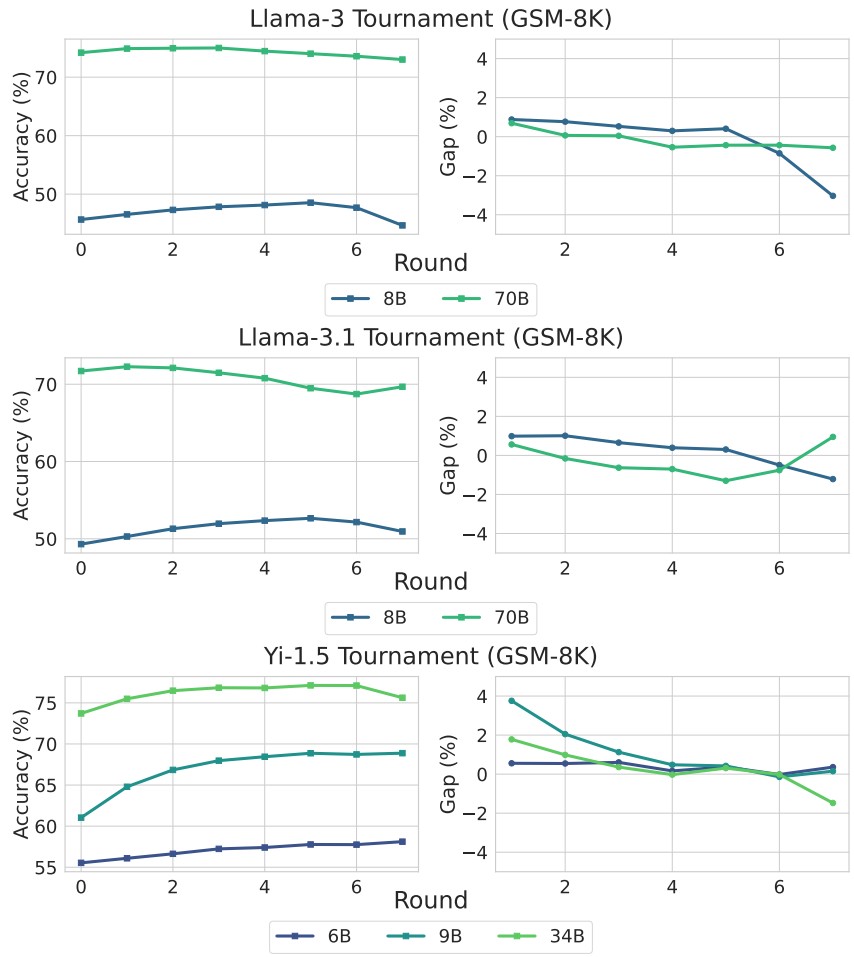

Figure 15: Change in the generation accuracy of the filtered dataset (left) and gap (right) with respect to the round of tournament. The right figure plots the gap with respect to the accuracy from the previous round instead of the base accuracy.

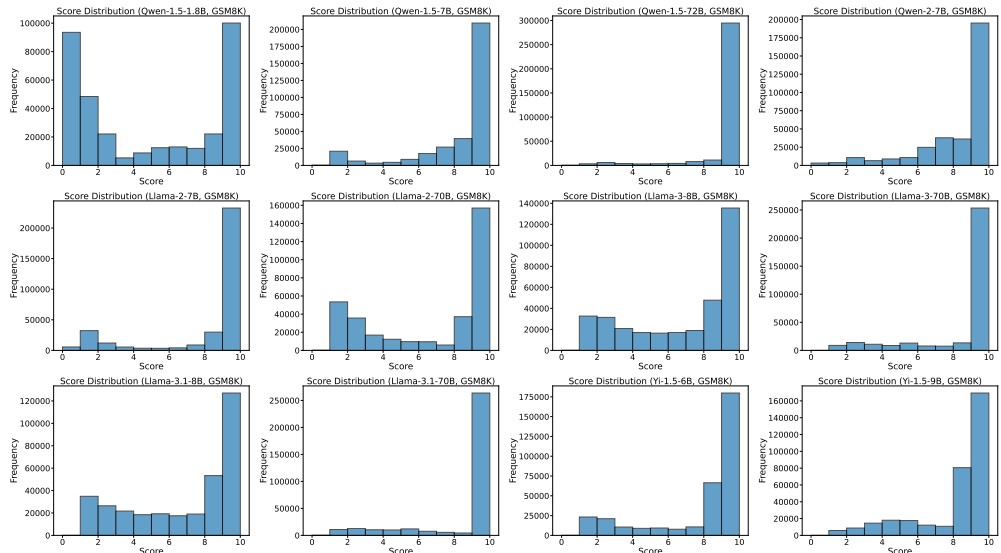

Figure 16: Score distribution of a subset of models. The mode of the score for all models is 10.

## E.6   ADDITIONAL RESULTS FOR SECTION 6.2

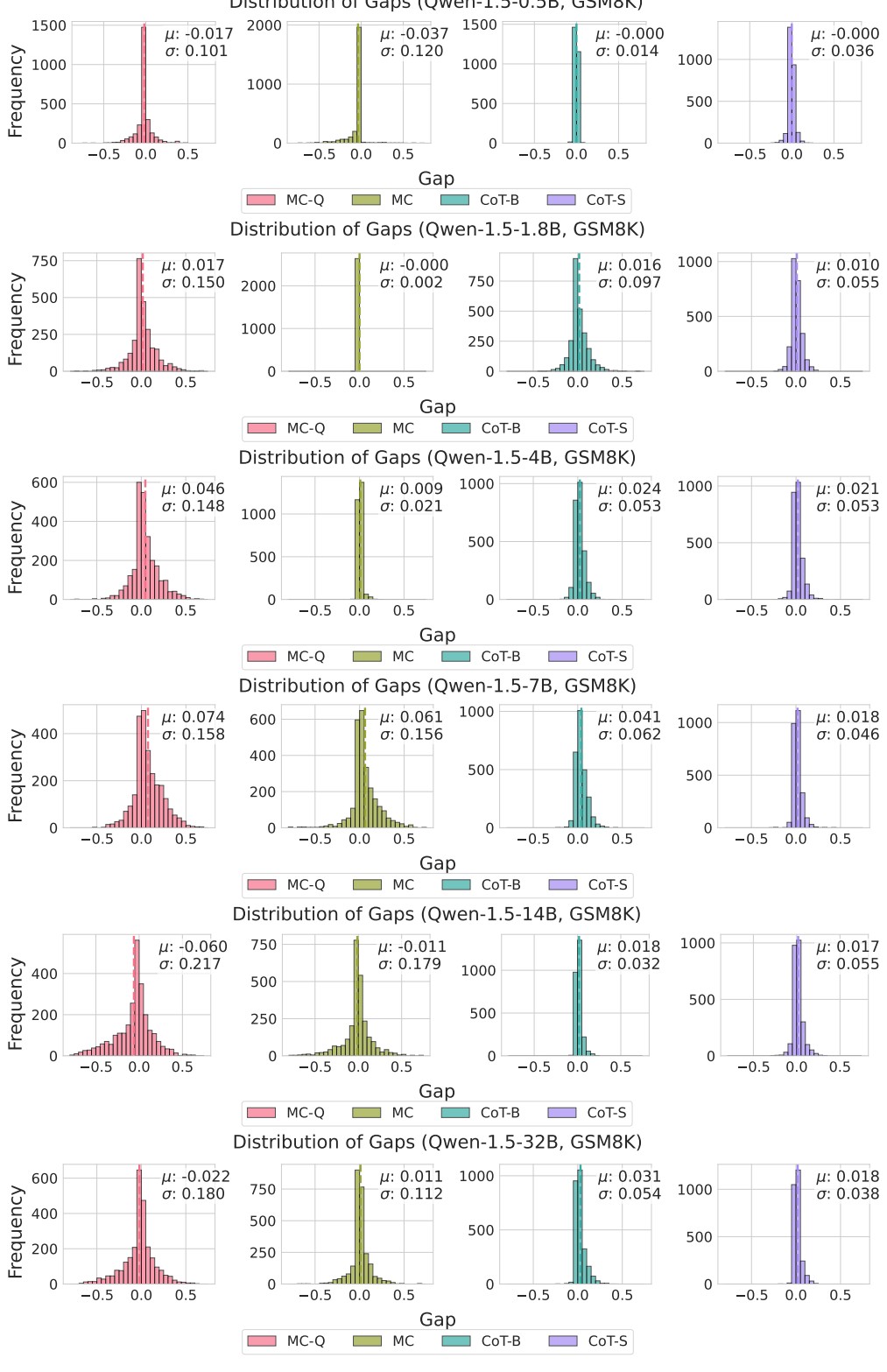

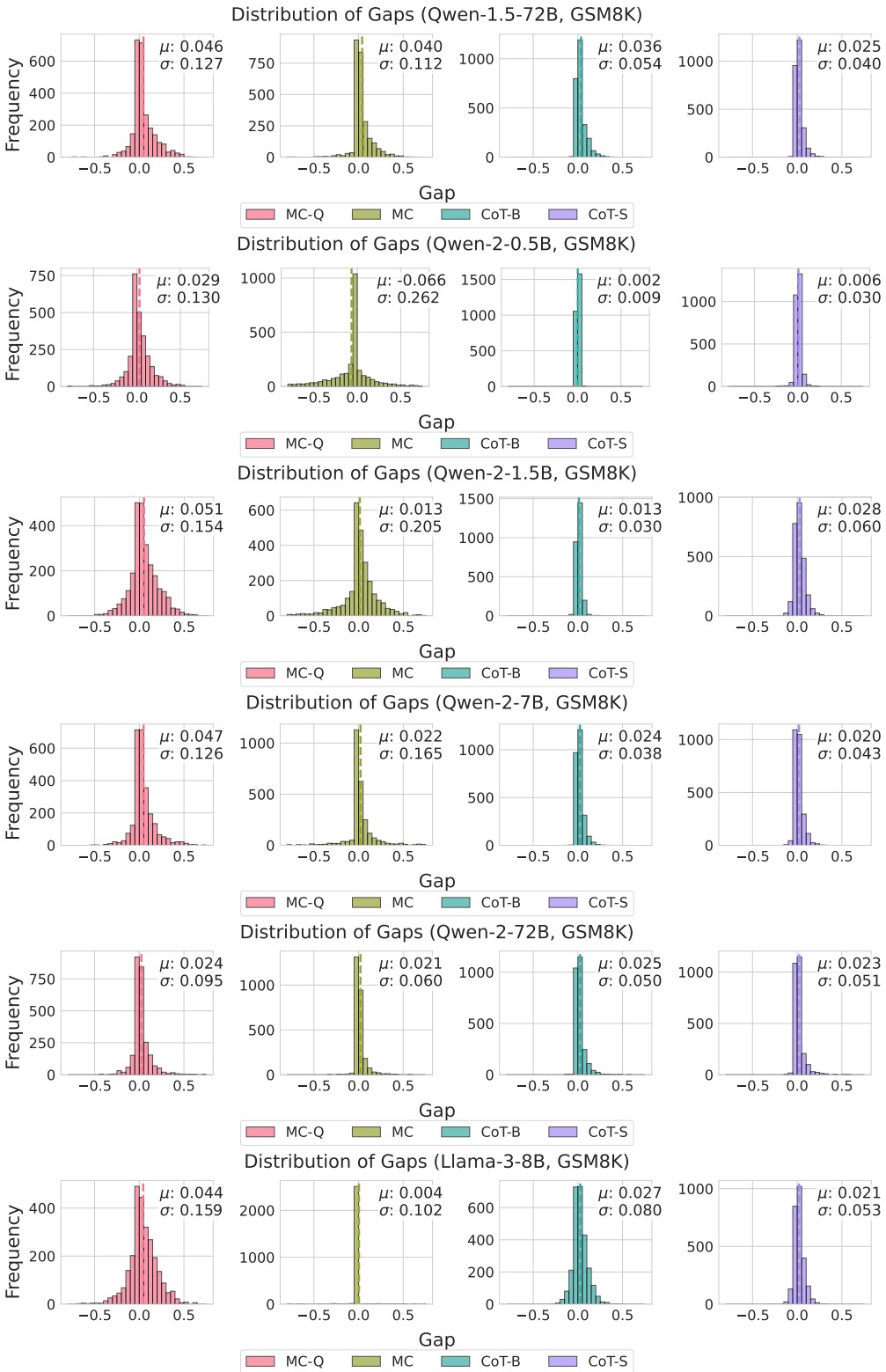

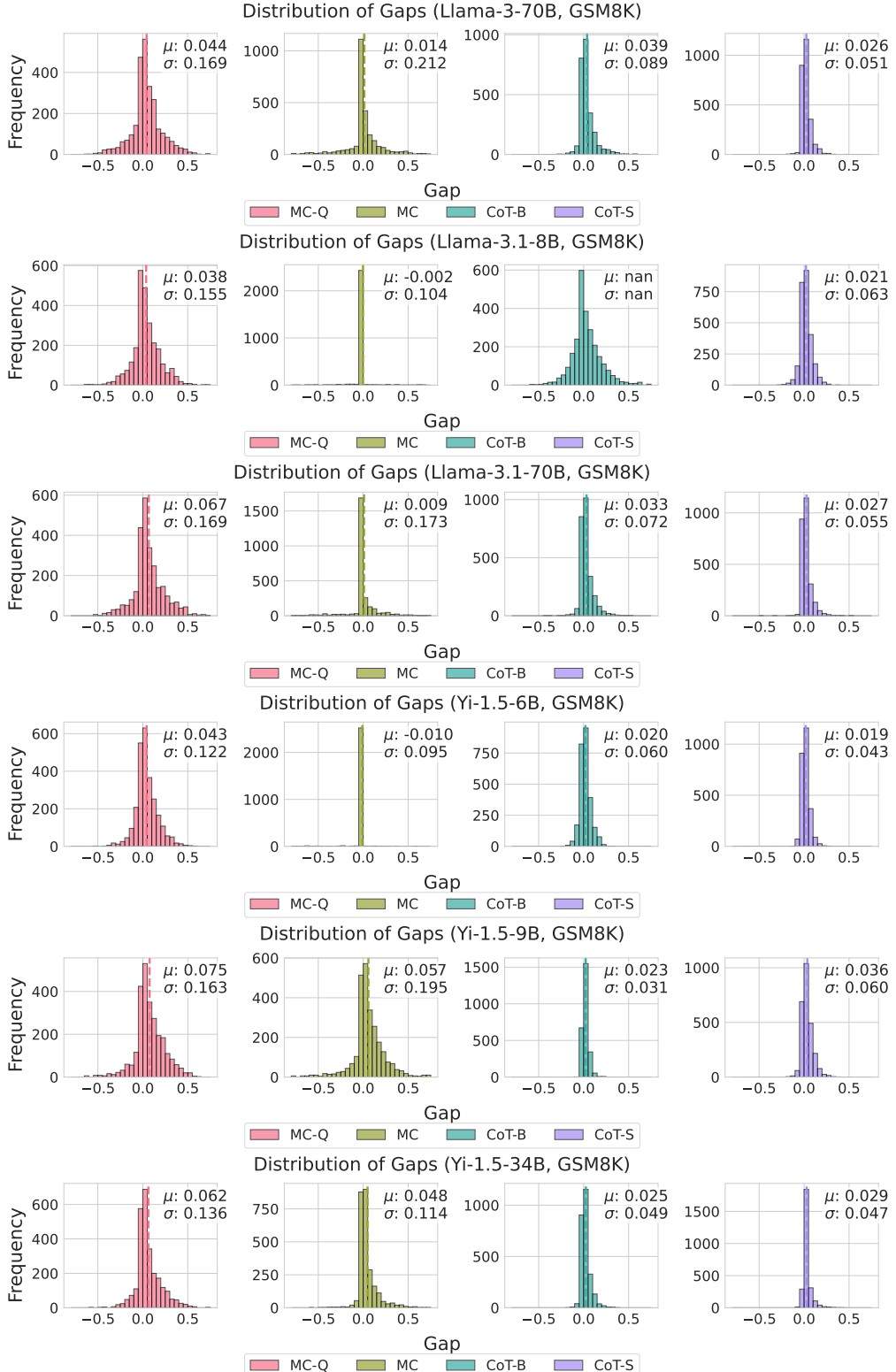

Figure 17: The empirical distribution of gaps of each verification method of each model on GSM8K. We cluster gaps in bins of intervals with width of 0.005. We label the mean ($\mu$) and standard deviation ($\sigma$) of each distribution.

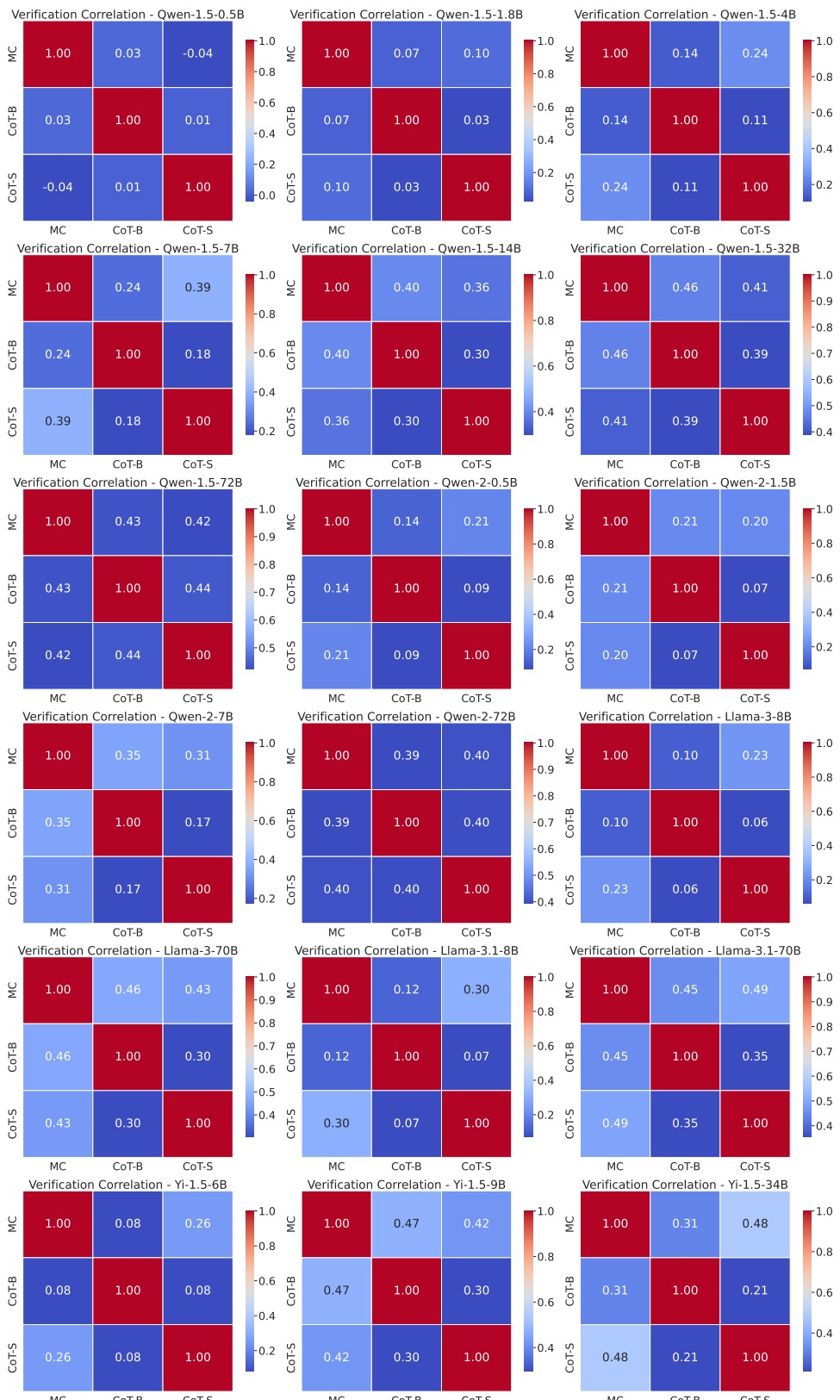

Figure 18: The correlation plot of the output of each verification $\widehat{u}$.

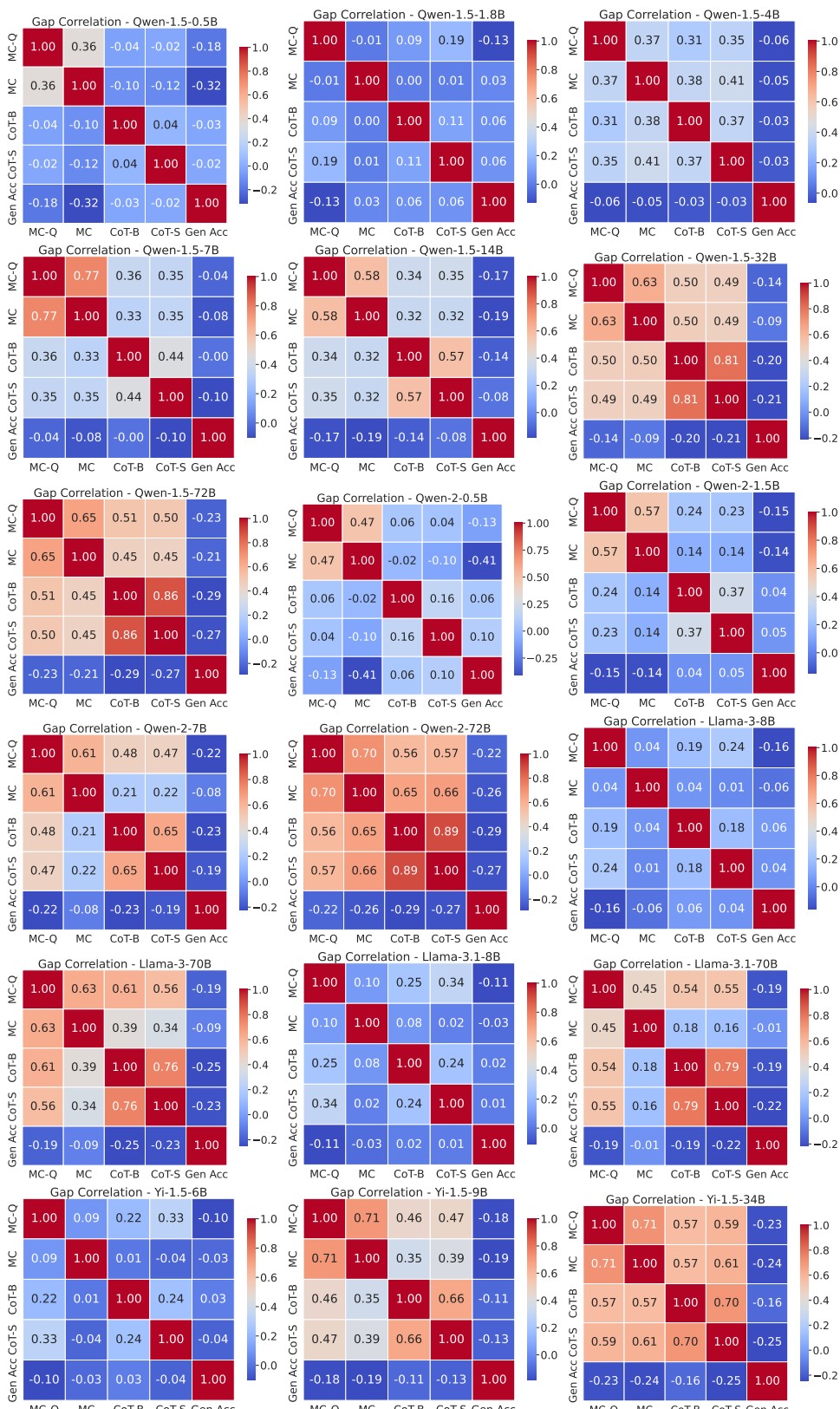

Figure 19: The correlation plot of the gap from each verification and generation accuracy.

### E.7 Additional Results for Section 6.3

Table 7: Relative gaps on GSM-8K for all models. For each verification, "top $n$" denotes taking the threshold as the $n$ quantile of the proxy utility for each prompt, and $\tau = n$ denotes taking the threshold as $n$ for all prompts. All numbers denote the percentage

| Name | Size | MC | CoT-B | CoT-S | MC+CoT-B | MC+CoT-S | CoT-B+CoT-S | All |
|---|---|---|---|---|---|---|---|---|
| Qwen-1.5 | 0.5B | -1.62 | -0.07 | -0.01 | -1.65 | -1.36 | -0.08 | -1.42 |
| | 1.8B | 2.36 | 1.04 | 0.95 | 2.16 | 2.22 | 1.18 | 2.14 |
| | 4B | 4.55 | 2.26 | 2.14 | 4.81 | 4.77 | 3.50 | 4.90 |
| | 7B | 7.46 | 3.98 | 1.84 | 9.02 | 7.88 | 4.88 | 9.12 |
| | 14B | -6.09 | 1.79 | 1.69 | -5.73 | -5.85 | 2.45 | -5.60 |
| | 32B | -2.30 | 3.07 | 1.84 | -1.87 | -2.21 | 3.62 | -2.00 |
| | 72B | 4.61 | 3.43 | 2.47 | 5.70 | 5.25 | 4.46 | 6.07 |
| Qwen-2 | 0.5B | 3.01 | 0.18 | 0.64 | 3.01 | 3.13 | 0.72 | 3.11 |
| | 1.5B | 5.04 | 0.86 | 2.78 | 5.08 | 7.68 | 3.15 | 7.61 |
| | 7B | 4.68 | 2.31 | 1.97 | 5.25 | 4.96 | 3.40 | 5.46 |
| | 72B | 2.44 | 2.50 | 2.28 | 3.21 | 3.13 | 3.35 | 3.65 |
| Llama-2 | 7B | 2.17 | 0.13 | 0.25 | 2.21 | 2.42 | 0.37 | 2.44 |
| | 13B | 3.19 | 0.91 | 0.97 | 3.47 | 3.38 | 1.76 | 3.21 |
| | 70B | 4.78 | 3.73 | 3.44 | 5.52 | 5.61 | 5.79 | 4.90 |
| Llama-3 | 8B | 4.98 | 2.59 | 2.10 | 5.88 | 5.81 | 4.09 | 5.83 |
| | 70B | 4.37 | 3.88 | 2.59 | 4.91 | 4.71 | 4.45 | 4.98 |
| Llama-3.1 | 8B | 4.34 | 3.50 | 2.09 | 4.47 | 4.57 | 3.27 | 2.96 |
| | 70B | 6.72 | 3.29 | 2.71 | 7.08 | 7.03 | 3.94 | 7.21 |
| Yi-1.5 | 6B | 4.27 | 2.01 | 1.88 | 4.83 | 4.72 | 3.16 | 4.95 |
| | 9B | 7.50 | 2.32 | 3.61 | 7.79 | 7.87 | 4.80 | 8.11 |
| | 34B | 6.23 | 2.49 | 2.86 | 6.32 | 6.35 | 3.76 | 6.41 |

# F    DETAILS ON ITERATIVE SELF-IMPROVEMENT

Here we present the details for RL update: we use Reasoning Preference Optimization (Pang et al., 2024), which is equivalent to the RL update under Bradley-Terry Model with an SFT regularization. We set the KL regularization coefficient $\beta$ to be 0.05 and the SFT regularization coefficient to be 1.

---

**Algorithm 1** Iterative self-improvement with rejection sampling.

---

**require**  Model class $\mathcal{F}$, prompt set $X$, threshold $\tau$.
1: Let $f_0$ be the pre-trained model.
2: **for** $t = 1, 2, \ldots, T$ **do**
3:     For each $x \in X$, sample $N$ generations $y \sim f_{t-1}(\cdot \mid x)$, and compute the score

$$s(x, y) = \widehat{u}_{f_{t-1}}(x, y).$$

4:     Get the filtered dataset $\mathcal{D}_t = \{(x, y) \mid s(x, y) > \tau\}$.
5:     Update the model through MLE to get $f_t$:

$$f_t = \arg\max_{f \in \mathcal{F}} \sum_{(x,y) \in \mathcal{D}_t} \log f(y \mid x).$$

---

Table 8: Hyperparameter for iterative self-improvement.

| | |
|---|---|
| Minibatch size | 64 |
| Learning rate | 1e-6 |
| Optimizer | AdamW |
| Gradient step | 2000 |
| Max Sequence Length | 2048 |
| Data Type | bf16 |

## G GENERATION AND VERIFICATION PROMPTS

---

**Multiple Choice Verification Prompt (GSM8K / nq_open)**

Judge the correctness of the following solution of the problem. Answer with either Correct or Incorrect. Problem: {problem}
Solution: {generation}
Judge:

---

**Chain of Thoughts Binary Prompt (GSM8K)**

Review the following math problem and the attempted solution and verify the correctness of the attempted solution, with a judgement of <correct> or <incorrect>. Your judgement should follow each criterion below: - The final ANSWER is after the phrase "The answer is ANSWER", and verify if the answer is correct with respect to the problem. If there is no such phrase, treat the answer as incorrect. - Each solution contains a derivation before the final answer, check the soundness of the derivation as well. - Your final judgement should reflect solely on the correctness of the final answer, but if there are issues in the derivation, please mention them in your justification.
Problem: {problem}
Attempted Solution: {generation}
After examining the problem and the attempted solution: - Briefly justify your judgement, up to 50 words. - Conclude with the judgement using the format: "Correctness: <correct> or <incorrect>".
Remember to assess from the math verifier perspective and be critical and verify carefully.
Judgement:

---

**Chain of Thoughts Score Prompt (GSM8K)**

Review the following math problem and the attempted solution and give a score from 1 to 10 to the attempted solution. The final ANSWER is after the phrase "Final Answer: The final answer is ANSWER". Give the answer a 1 if there is no such phrase or ANSWER is wrong, and give the answer a 10 if both the answer and the derivation are correct.
Problem: {problem}
Attempted Solution: {generation}
After examining the problem and the attepmted solution: - Briefly justify your score, up to 50 words. - Conclude with the score using the format: "Score: <score>".
Remember to assess from the math verifier perspective and be critical and verify carefully.
Judgement:

---

**Tournament Prompt (GSM8K)**

Review the following math problem and two attempted solutions. Your task is to determine the better solution between the two. Your judgement should follow each criterion below: - The final ANSWER is after the phrase "The answer is ANSWER", and you should always prefer correct answers over incorrect answer. - Always prefer solutions with the phrase "The answer is ANSWER" over ones without it. - If both answers are correct or incorrect, you should prefer the one with better reasonings. Problem: {problem}
Solution A: {generation1}
Solution B: {generation2}
After examining the problem and the attempted solutions: - Briefly justify your judgement, up to 50 words. - Conclude with the judgement using the format: "Preferred solution: <A> or ".
Remember to assess from the math verifier perspective and be critical and verify carefully.
Judgement:

---

---

**Sudoku Generation Prompt**

You are a Sudoku solver specialized in 4x4 puzzles. You will be given a string of 16 digits representing an initial 4x4 Sudoku puzzle, where 0 represents an empty cell. Your task is to solve the puzzle and provide the complete solution.

Rules for solving a 4x4 Sudoku:

Each row must contain the numbers 1-4 without repetition.

Each column must contain the numbers 1-4 without repetition.

Each 2x2 quadrant must contain the numbers 1-4 without repetition.

The solution must maintain all the initial non-zero numbers from the puzzle in their original position.

Input format:

ab cd

ef gh

ij kl

mn op

Where each letter represents a single digit (0-4), and 0 indicates an empty cell. Your tasks:

Analyze the given puzzle.

Fill in all empty cells (0s) with appropriate numbers following Sudoku rules.

Provide your solution in the same format as the input, maintaining the original structure.

Puzzle: {problem}

Answer:

---

**Sudoku Generation Example**

Puzzle:

03 21

00 30

04 00

21 00

Answer: I filled A1 with 4 to complete Row 1. In Quadrant 1, the missing numbers 1 and 2 were assigned to B1 and B2 based on Column constraints. B4 was set to 4 to complete Row 2. In Quadrant 3, C1 was filled with 3. Row 3 missing numbers 1 and 2 were placed in C3 and C4. D4 was assigned 3 to complete Column 4, and D3 was set to 4 to finish Row 4.

Solution:

43 21

12 34

34 12

21 43

---

---

**Sudoku Verification Prompt**

You are a Sudoku validator for 4x4 puzzles. You will be given two strings of 16 numbers each: the first represents the initial Sudoku puzzle, and the second represents the proposed solution. Your task is to determine if the solution is correct.
Rules for a valid 4x4 Sudoku solution:
Each row must contain the numbers 1-4 without repetition.
Each column must contain the numbers 1-4 without repetition.
Each 2x2 quadrant must contain the numbers 1-4 without repetition.
The solution must maintain all the initial numbers from the puzzle.
Input format:
Question: Input format:
ab cd
ef gh
ij kl
mn op
Where each letter represents a single digit (0-4), and 0 indicates an empty cell. Answer: [same format but there should be no 0]
Your task:
Check if the solution maintains all initial numbers from the puzzle.
Verify that each row, column, and 2x2 quadrant contains 1-4 without repetition.
Respond with "Correct" if the solution is valid, or "Incorrect" if it's not.
Question: {problem}
Answer: {generation}
Judgment:

---

**Sudoku Verification Example**

problem:
03 00
21 00
00 01
10 43
generation:
34 12
21 31
34 21
12 44
correctness: Checking if the solution maintains all initial numbers from the puzzle:
Position (Row 1, Column 2): Puzzle has 3, Solution has 3 — Correct
Position (Row 2, Column 1): Puzzle has 2, Solution has 2 — Correct
Position (Row 2, Column 2): Puzzle has 1, Solution has 1 — Correct
Position (Row 3, Column 4): Puzzle has 1, Solution has 1 — Correct
Position (Row 4, Column 1): Puzzle has 1, Solution has 1 — Correct
Position (Row 4, Column 3): Puzzle has 4, Solution has 4 — Correct
Position (Row 4, Column 4): Puzzle has 3, Solution has 3 — Correct
All initial numbers are maintained.
Checking the rows:
Row 1: Numbers are 4, 3, 1, 2 (No repetition) - Correct
Row 2: Numbers are 2, 1, 3, 1 (Number 1 repeats) - Invalid
Row 3: Numbers are 3, 4, 2, 1 (No repetition) - Correct
Row 4: Numbers are 1, 2, 4, 3 (No repetition) - Correct
Checking the columns:
Column 1: Numbers are 4, 2, 3, 1 (No repetition) - Correct
Column 2: Numbers are 3, 1, 4, 2 (No repetition) - Correct
Column 3: Numbers are 1, 3, 2, 4 (No repetition) - Correct
Column 4: Numbers are 2, 1, 1, 3 (Number 1 repeats) - Invalid

Checking the 2x2 quadrants:
Quadrant 1 (Rows 1-2, Columns 1-2): Numbers are 4, 3, 2, 1 (No repetition) - Correct
Quadrant 2 (Rows 1-2, Columns 3-4): Numbers are 1, 2, 3, 1 (Number 1 repeats) - Invalid
Quadrant 3 (Rows 3-4, Columns 1-2): Numbers are 3, 4, 1, 2 (No repetition) - Correct
Quadrant 4 (Rows 3-4, Columns 3-4): Numbers are 2, 1, 4, 3 (No repetition) - Correct
Conclusion: There are repetitions in Row 2, Column 4, and Quadrant 2, the solution violates the Sudoku rules.
Therefore, the response is: Incorrect

## H  INFRASTRUCTURE STATEMENT

All our inferences are performed on a cluster of Nvidia A100 40GiB nodes, and our iterative self-improvement training experiments are performed on a cluster of Nvidia A100 80GiB nodes.

