# OpenReview forum: "Mind the Gap: Examining the Self-Improvement Capabilities of Large Language Models"
_ICLR.cc/2025/Conference — ICLR 2025 Oral_

### Official Review · Reviewer_1Q5B · 2024-11-01

**Soundness:** 3
**Presentation:** 3
**Contribution:** 3
**Rating:** 6
**Confidence:** 4

**Summary:**

This paper studies the self-improvement phenomenon of LLMs. The authors have proposed a mathematical framework to formally discuss the LLM self-improving process, and introduced a well-motivated metric called generation-verification gap (GV-gap) for observation. Linear scaling properties between the GV-gap and pretraining FLOPs are illustrated. The authors have adopted three types of distinct tasks for a comprehensive study of the LLM self-improvement process.

**Strengths:**

- The authors conducted a timely study on self-improvement of LLMs.

- The proposed mathematical framework about LLM self-improvement is insightful, especially the design of the generation-verification gap (and its relative variant).

- The designed experiments (GSM-8K, NQ, and Sudoku) are representative, as they correspond with the settings of the improvable, the unimprovable, and the "theoretically improvable, but empirical findings show unimprovable".

- The scaling property of GV-gap ~ pretraining FLOPs is a nice read.

**Weaknesses:**

Please see the Questions below.

**Questions:**

- Is the RL update illustrated in Example 2.1 adopted in experiments?

- Does the **High-fidelity model update** condition mentioned in Section 2.1 examined / verified in experiments？What's the practical benefit for having this condition (principle)?

- Will different decoding algorithms (e.g., search algorithms) for generation help LLMs to (self-)improve? And will the choices of different generation methods alter the conclusions drawn in this paper?

- I like the experiments with Sudoku (Section 4.4), as this is a setting with answer generation strictly harder than verification (NP-hard > P; assuming NP!=P). However, according to the experimental results, non-trivial gaps are observed only with models of > 70B parameters. I wonder if the authors could conduct additional experiments by sweeping the settings of the Sudoku problems: currently it's of 4x4 matrices; how about 3x3, or even 2x2 sized matrices, where the problems should be much easier? With these experiments, can we have the observation that LLMs with different sizes can self-improve?

- Can the authors **use the mathematical framework proposed in Section 2** to provide insights about why the diversity deteriorates with iterative rejection fine-tuning?

- Further thoughts: if diversity deteriorates, do the authors mean that the self-improving process is in fact harmful in some aspect to LLMs, and eventually the self-improvement is just about some kind of tradeoff? What's the authors' opinion on this?

---

> ### Author Response · Authors · 2024-11-21
> **Response (1/2)**
>
> We thank the reviewer for their insightful questions. Please see responses to individual comments below. We hope to address your concerns and we look forward to a constructive discussion.
>
> > Is the RL update illustrated in Example 2.1 adopted in experiments?
>
> In the revised submission, we added the RL update result in revised Fig. 4 (now highlighted in blue). The RL update is labeled as “7B-RL” with a star marker on the plot. We observe that RL update has very similar dynamics as rejection sampling. We appreciate the question and we believe the RL result will improve the completeness of our paper.
>
> > Does the High-fidelity model update condition mentioned in Section 2.1 examined / verified in experiments？What's the practical benefit for having this condition (principle)?
>
> The high-fidelity model update condition can be examined implicitly in Section 5 on the left of figure 4 by comparing the gap and performance difference in each iteration. To make this examination more explicit, below we provide the Pearson correlation coefficient between the gap and actual performance different across all models and verification method in our GSM8K experiments:
>
> | Iteration    | Value  |
> |--------------|--------|
> | Iteration 1  | 0.643  |
> | Iteration 2  | 0.783  |
> | Iteration 3  | 0.918  |
>
> We believe this analysis exactly captures the motivation of using gap to measure the self-improvement capabilities of LLMs - while gap and final performance always have strong positive correlation, it is lower in the initial iterations because confounders are introduced during early model update.
>
> High-fidelity model update is important in the iterative self-improvement setting, because we need a distillation of the filtered dataset (which has higher accuracy than the original model) so that we will have an improved generator for the next iteration.
>
> > Will different decoding algorithms (e.g., search algorithms) for generation help LLMs to (self-)improve? And will the choices of different generation methods alter the conclusions drawn in this paper?
>
> We believe using search algorithms such as MCTS can indeed improve as shown in the inference-time scaling papers [1,2] (although during search they rely on a stronger reward model which is absent in the self-improvement framework). However, it does not strictly fit into our framework as searching algorithms are using a different generation distribution other than the base model, and performing such experiments requires another order of magnitude of compute so we will leave it as future work. Previous works on scaling laws in downstream tasks also only samples from the models without any searching algorithms [3].
>
> That said, in Appendix B.1, we record the results for Qwen-1.5 model family in GSM8K, with different sampling temperatures. We observe the similar results as long as the temperature is within a reasonable range. We hope this result can help demonstrate the robustness of our result under different sampling conditions.
>
> > I like the experiments with Sudoku (Section 4.4), as this is a setting with answer generation strictly harder than verification (NP-hard > P; assuming NP!=P). However, according to the experimental results, non-trivial gaps are observed only with models of > 70B parameters. I wonder if the authors could conduct additional experiments by sweeping the settings of the Sudoku problems: currently it's of 4x4 matrices; how about 3x3, or even 2x2 sized matrices, where the problems should be much easier? With these experiments, can we have the observation that LLMs with different sizes can self-improve?
>
> Thank you for your appreciation of our sudoku result! We agree with the reviewer that trying an easier P vs. NP problem is of great interest. However, sudoku can only be of size of $n^2 \times n^2$, so a smaller sudoku puzzle is 1x1 and the problem becomes trivial. We remark that we also tried the canonical 9x9 sudoku but it is even challenging for capable closed-sourced models such as Sonnet 3.5 or ChatGPT 4(o). That said, we are excited to try other tasks in the future.

---

> > ### Author Response · Authors · 2024-11-21
> > **Response (2/2)**
> >
> > > Can the authors use the mathematical framework proposed in Section 2 to provide insights about why the diversity deteriorates with iterative rejection fine-tuning?
> >
> > We believe this is a great question and we believe fully understanding the underlying dynamics of this interesting phenomenon would be an interesting direction for future research. Here we provide an intuitive answer based on our mathematical framework. Fix a model $f$ to be the both generator and verifier, and one can believe that the density of a response $y$ is positively correlated with the proxy utility, i.e., $f(y)$ and $\hat u_f(y)$ are positively correlated in the support of $f(y)$, then the filtered distribution is $f(y) \cdot w(\hat u_f(y)) / Z$, where $Z$ is the normalization factor.  We can see that the density is even more concentrated on the generations that already have high density. Thus assuming that the updated model can have similar distribution and the filtered dataset (the high-fidelity update condition), then we have the next iteration model $f’ = f(y) \cdot w(\hat u_f(y)) / Z$ in distribution. Finally, we know that the verifier is not perfect, so there are responses $y$ which with high $\hat u_f(y)$ but low ground truth utility $u(y)$ (e.g., a wrong generation). Thus $f’$ will have lower diversity compared to $f$, because on some $x$, $f’$ is more concentrated on a wrong generation.
> >
> > To complement this result, in Appendix B.3, we show that if we use ground truth label as the verifier (such that no wrong answer will have increasing density), we do not observe the decrease in effective diversity, which agrees with the theoretical explanation.
> >
> > > Further thoughts: if diversity deteriorates, do the authors mean that the self-improving process is in fact harmful in some aspect to LLMs, and eventually the self-improvement is just about some kind of tradeoff? What's the authors' opinion on this?
> >
> > We think whether self-improvement is harmful or not depends on the metric. In practice, we believe that the most important metric is still the final performance (e.g., accuracy) so in general, self-improvement is still helpful. In certain situations it also makes sense to care about the diversity, and in this case we can design a proxy utility that promotes the diversity of the generation (e.g., with prompt to give a high score for diverse generation or simply use empirical entropy), and we can have a self-improvement loop that increases the diversity along the iterations. One can also achieve certain tradeoffs by combining the aforementioned utilities for both accuracy and diversity.
> >
> > Overall we think this is an interesting question, and as we mentioned throughout the paper, how to self-improve performance without sacrificing diversity is an interesting future direction. We will make sure to emphasize this discussion in the paper.
> >
> > ### Reference
> >
> > [1] Yangzhen Wu, Zhiqing Sun, Shanda Li, Sean Welleck, and Yiming Yang. An empirical analy-
> > sis of compute-optimal inference for problem-solving with language models. arXiv preprint
> > arXiv:2408.00724, 2024.
> >
> > [2] Charlie Snell, Jaehoon Lee, Kelvin Xu, and Aviral Kumar. Scaling llm test-time compute optimally can be more effective than scaling model parameters. arXiv preprint arXiv:2408.03314, 2024.
> >
> > [3] Isik, Berivan, et al. "Scaling laws for downstream task performance of large language models." arXiv preprint arXiv:2402.04177 (2024).

---

> > > ### Comment · Reviewer_1Q5B · 2024-11-22
> > >
> > > Thanks for the authors' detailed responses to my questions. The contributions of this work are clearer to me now. I raised my rating to 6.

---

> > > > ### Author Response · Authors · 2024-11-22
> > > >
> > > > We are glad to see that our rebuttal helps clarify the contribution and we thank the reviewer for their timely reponse!

---

### Official Review · Reviewer_gkZY · 2024-11-02

**Soundness:** 3
**Presentation:** 3
**Contribution:** 3
**Rating:** 8
**Confidence:** 3

**Summary:**

The paper provides a mathematical formulation of LLM self-improvement and defines generation-verification gap to measure the potential of self-improvement. Based on the metric, the work conducts extensive experiments in terms of scaling laws, tasks suitable for self-improvement, iterative procedure and better verification attempts.

**Strengths:**

1. The paper presents a math formulation and a new metric for studying self-improvement, which I believe will be beneficial for current studies related to LLM self-improvement.
2. The experiments are comprehensive, analyzing self-improvement from various perspective.
3. The experimental findings are generally insightful.
4. The paper is well-written and easy to follow.

**Weaknesses:**

1. My main concern is about the experiment setup. The main experiments are limited to one benchmark, GSM8k, in the math task. Despite GSM8k is a classical benchmark for mathematics, it is relatively easy for current cutting-edge LLMs and its benchmarking function may decrease due to potential data contamination. Additionally, the self-improvement performance as well as the findings may differ across different datasets or scenarios. Therefore, I think it would be beneficial for the authors to complement experiments on more datasets and scenarios.
2. The defined metric reveals the capability gap between generation and verification. Empirically the metric is useful in predicting potential self-improvement. But it would be more solid if the authors could analyze the correlations between the metric and the actual performance changes after iterations of self-improvement. Experimental evidences and analyses would provide more understanding of the metric and its potential use.

**Questions:**

1. The findings on "effective diversity of generations" are interesting. Have you conducted further experiments to clarify the underlying causes? For instance, could it be attributed to the training process itself? (Perhaps an experiment involving training on golden labels could provide insights.) Alternatively, could it stem from noise introduced by self-verification? (An experiment to filter out incorrect samples using golden labels or employing stronger evaluators might help determine if this mitigates the issue.)
2. From the main results on scaling law (Figure 1), the scaling laws seem to be severely influenced by the verification methods. Why do some methods (like MC, To) not exhibit such scaling laws? And whether the other methods share similar findings (like CoT-B in the paper and some others)?

---

> ### Author Response · Authors · 2024-11-21
> **Response**
>
> We thank the reviewer for their insightful feedback. Please see responses to individual questions below. We hope to address your concerns and we look forward to a constructive discussion.
>
> > My main concern is about the experiment setup. The main experiments are limited to one benchmark, GSM8k, in the math task. Despite GSM8k is a classical benchmark for mathematics, it is relatively easy for current cutting-edge LLMs and its benchmarking function may decrease due to potential data contamination. Additionally, the self-improvement performance as well as the findings may differ across different datasets or scenarios. Therefore, I think it would be beneficial for the authors to complement experiments on more datasets and scenarios.
>
> Thank you for pointing this out. We want to first remark that we indeed examine self-improvement on information retrieval tasks in Section 4.3 and Sudoku in Section 4.4. We hope these results can provide more completeness to our overall message.
>
> On the other hand, we are excited to share our new results on the more challenging MATH dataset [1], which consists of competition level math questions. We repeat the same experiments as in GSM8K on all model families, and an additional Qwen-2.5 model family that consists of 7 models. We observed similar results in both scaling experiments and iterative self-improvement experiments. We present the results in Appendix A in the revised submission.
>
> > The defined metric reveals the capability gap between generation and verification. Empirically the metric is useful in predicting potential self-improvement. But it would be more solid if the authors could analyze the correlations between the metric and the actual performance changes after iterations of self-improvement. Experimental evidences and analyses would provide more understanding of the metric and its potential use.
>
> Thank you for the suggestion. Below we present the Pearson correlation coefficient between the gap and actual performance different across all models and verification method in our GSM8K experiments:
>
> | Iteration    | Value  |
> |--------------|--------|
> | Iteration 1  | 0.643  |
> | Iteration 2  | 0.783  |
> | Iteration 3  | 0.918  |
>
> We believe this analysis exactly captures the motivation of using gap to capture the self-improvement capabilities of LLMs - while gap and final performance always have strong positive correlation, it is lower in the initial iterations because confounders are introduced during early model update. For example, as we discussed in the paper, models might be just better at following format instead of better at solving the problems, and thus the final performance gap might be higher than the gap in the smaller models. Similar observations are present in other concurrent papers [2], and we argue that directly measuring performance difference is not the ideal metric. We hope the results can better motivate that gap is the more rigorous metric.
>
> > The findings on "effective diversity of generations" are interesting. Have you conducted further experiments to clarify the underlying causes? For instance, could it be attributed to the training process itself? (Perhaps an experiment involving training on golden labels could provide insights.) Alternatively, could it stem from noise introduced by self-verification? (An experiment to filter out incorrect samples using golden labels or employing stronger evaluators might help determine if this mitigates the issue.)
>
> Thank you for the suggestion. We conduct additional experiments as suggested by the reviewer, with iterative improvement on Qwen-1.5-7B model on GSM8K in the following two settings: 1) using gold labels and 2) use Qwen-1.5-72B model as the verifier. We observe that there is no decrease in effective diversity if we use the gold label as verifier, and the decrease in effective diversity still happens in the second setting. This suggests that one cause of this phenomenon is that the model concentrates on some wrong answers that the imperfect verifier labels as correct. We refer the reviewer to Appendix B.3 for the concrete results.
>
> > From the main results on scaling law (Figure 1), the scaling laws seem to be severely influenced by the verification methods. Why do some methods (like MC, To) not exhibit such scaling laws? And whether the other methods share similar findings (like CoT-B in the paper and some others)?
>
> We believe one reason is that MC and To have higher variance in terms of gap across prompts as we show in section 6.1, which prevents a consistent trend due to the noise. For CoT-B, we present the results in Table 5 in the appendix, and we can see it has monotone scaling except the Qwen-1.5-14B model. For a more positive result, we observe a more consistent pattern in our new results under the MATH dataset (Appendix A), where CoT-B, CoT-S, and even MC demonstrate the scaling property.

---

> > ### Author Response · Authors · 2024-11-21
> > **Reference**
> >
> > [1] Dan Hendrycks, Collin Burns, Saurav Kadavath, Akul Arora, Steven Basart, Eric Tang, Dawn Song, and Jacob Steinhardt. Measuring mathematical problem solving with the math dataset. arXiv preprint arXiv:2103.03874, 2021
> >
> > [2] Xuanchang Zhang, Wei Xiong, Lichang Chen, Tianyi Zhou, Heng Huang, and Tong Zhang. From lists to emojis: How format bias affects model alignment. arXiv preprint arXiv:2409.11704, 2024.

---

> > ### Comment · Reviewer_gkZY · 2024-11-22
> >
> > Thanks for the detailed response! The additional experiments have largely addressed my concerns and make this work more solid. I believe this work has the potential to be impactful to LLM self-improvement research and I decided to raise my score to 8.

---

> > > ### Author Response · Authors · 2024-11-22
> > >
> > > We appreciate the reviewer's recognition of the additional experiments and the positive evaluation of the potential of our work!

---

### Official Review · Reviewer_QqKc · 2024-11-03

**Soundness:** 3
**Presentation:** 3
**Contribution:** 3
**Rating:** 8
**Confidence:** 4

**Summary:**

This work performs an extensive and exhaustive analysis on self-improvement abilities of large language models. First, authors present related work under which the self-improvements capabilities has been studied and formulate a high-level, unified way of representing such framework. Then they propose a novel way of measuring the self-improvement capability using the so called generation-verification gap (GV-gap). Further, they design an experimental setup and provide multiple takeaways that helps to bring the light onto the cases where self-improvement is likely very limited and hypothesize why this might be happening. Finally, authors provided multiple future directions based on their takeaways and findings.

**Strengths:**

* The proposed measure (gv-gap) is clearly defined with real world examples when it comes to different cases of generators and verifiers. This brings good clarify into the overall presentation.
* The view on self-improvements via the proposed gap brings a new way to quantify the limits of self-improvements, that is a contribution into this research direction and can be used by the research community in future work.

**Weaknesses:**

* While the definition of the gv-gap is pretty strict, the real world choice of the utility function making the practical measurements very noisy as authors pointed out in the text (which is great!). They have proposed some ways on improving the verification signal, however, that is a weak point in the entire framework that does not present itself when we just measure the model performance before and after self-improvement steps.
* While I agree that studying base models makes things much easier to handle in paper-size experimental setup, the absence of any signal about the gap changes with instruct based model weakens the provided takeaways. For instance, recent work showed that smaller (8B) instruct models are self-improving very well while in this work there is an opposite signal from the base smaller models that are seemingly lacking reasoning capabilities even under few shot settings.

**Questions:**

* adding more about the limitations coming from only using the base models would make overall takeaways not weaker, but clearer w.r.t. other recent research in this area.
* overall I think this will be a very useful analysis work for future self-improvement research direction.

---

> ### Author Response · Authors · 2024-11-21
> **Response**
>
> We thank the reviewer for their insightful comments. Please see responses to individual questions below. We hope to address your concerns and we look forward to a constructive discussion.
>
> > While the definition of the gv-gap is pretty strict, the real world choice of the utility function making the practical measurements very noisy as authors pointed out in the text (which is great!). They have proposed some ways on improving the verification signal, however, that is a weak point in the entire framework that does not present itself when we just measure the model performance before and after self-improvement steps.
>
> Thank you for acknowledging our remark on the noisiness of the verification procedure. We agree very much that the current approach for verification is not ideal, so we try to understand it in both theoretical and empirical ways. In theory, one can reduce the noise of the estimation of the gap by either sampling more verifications or by simply generating more candidate responses. If given a fixed compute budget, there will be a tradeoff between making more generations or making more verification, depending on the variance of each process. We think this is actually an interesting future direction with good practical implications.
>
> To have a better understanding on the variance of the verification, we perform the following experiment (detailed in Appendix B.4): with the Qwen-1.5B-72B model on GSM8K, we sample 8 generations for each question, and for each generation we sample 32 verification (CoT-B), and measure the empirical variance of the 32 verifications for each generation. We observe that the variance of the verification is near 0 for most of the generations, which indicates that in practice a handful of samples of verifications might suffice for an accurate estimation of the gap. We hope this result will make the discussion we had on the noisiness of the verification more complete.
>
> > While I agree that studying base models makes things much easier to handle in paper-size experimental setup, the absence of any signal about the gap changes with instruct based model weakens the provided takeaways. For instance, recent work showed that smaller (8B) instruct models are self-improving very well while in this work there is an opposite signal from the base smaller models that are seemingly lacking reasoning capabilities even under few shot settings.
>
> > adding more about the limitations coming from only using the base models would make overall takeaways not weaker, but clearer w.r.t. other recent research in this area.
>
> Thank you for the suggestion! We agree that analyzing instruct models is more aligned with the recent research in the area. As the reviewer mentions, finetuned models have more confounders that may not lead to a clean conclusion. Another reason that we were hesitant to perform the instructed model experiment is that some latest models such as llama 3.1 already have a self-improvement component in the finetune process so it is unclear if we will see any signal with such models and may confound various analyses. We will make sure to add more discussion in the paper.
>
> Towards a better understanding, we conduct the following experiments on instruct models: Qwen-1.5-Chat on GSM8K and Qwen-2.5-Instruct on MATH. We observe that Qwen-1.5-Chat models do not have the scaling property. In fact, the accuracy does not even increase monotonically with respect to the model size (and our accuracy result matches the reported results in the technical report). Also, some instruct model has lower accuracy than the base model. On the other hand, Qwen-2.5-Instruct models indeed have the scaling property. In this sense, we think the new result provides completeness as the reviewer suggests but also motivates our focus on the base models as well, since the instruct models introduce more confounders.
>
> > overall I think this will be a very useful analysis work for future self-improvement research direction.
>
> Thank you for acknowledging our contributions and we hope our work can benefit the community as you suggested.

---

### Official Review · Reviewer_eq2C · 2024-11-05

**Soundness:** 3
**Presentation:** 3
**Contribution:** 3
**Rating:** 6
**Confidence:** 4

**Summary:**

The paper investigates the concept of self-improvement in Large Language Models (LLMs) during pre-training, post-training, and test-time inference. The authors present a framework where the model verifies its own outputs, refines or reweights data based on this verification, and distills the filtered data. Despite empirical successes, there is a lack of fundamental understanding of this process. The study offers a comprehensive, modular, and controlled exploration of LLM self-improvement, introducing a mathematical formulation governed by the "generation-verification gap." Through experiments across different model families and tasks, the authors identify a scaling phenomenon linked to this gap that scales with model pre-training computational efforts. Additionally, the paper examines conditions under which self-improvement is feasible, proposes an iterative procedure for self-improvement, and suggests methods to enhance its performance. The findings not only deepen the understanding of LLM self-improvement but also open new research possibilities regarding its capabilities and limitations.

**Strengths:**

1. The research focus of this paper is commendable, as the GV-Gap aids in determining whether a model possesses self-improvement capabilities at any stage (Pre-/Post-training).

2. The overall layout are well-organized.

3. The experiments appear to be extensive and comprehensive. (Respect to Your Hard working)

**Weaknesses:**

### Deficiencies in Expression

1. The paper lacks a clear definition of the Generation-Verification Gap, with no illustrative diagram and only complex, confusing formulas cluttered with symbols.

2. The authors fail to clearly outline the overall methodological framework or how the experimental process operates. (I have read the experiments three times and even couldn't know how to evaluate the experimental performance! 🤷🏻‍♀️)

### Limitations in Contribution

The contributions to the community are relatively weak, as most of the conclusions are already known, such as the "verification gap scales monotonically with the model pre-training flops" and the "Saturation Limit and Cause of Saturation."
As a purely experimental analysis paper, it does not offer new insights, which is regrettably disappointing.

### Experimental Deficiencies
1. The experiments are primarily based on GSMK8k, which consists only of elementary-level math problems, making the results less convincing.
2. In Section 3, the experimental setup has predetermined parameters for lm-evaluation-harness tasks. However, it is unclear why so many hyperparameters would affect the GV-Gap instead of being solely determined by differences between the verifier and the generator.

**Questions:**

Please Check Weakness.

**Details Of Ethics Concerns:**

There are no ethical concerns about this paper.

---

> ### Author Response · Authors · 2024-11-21
> **Reponse (1/2)**
>
> We thank the reviewer for their careful review. Please see responses to individual questions below. We hope to address your comments and we look forward to a constructive discussion.
>
> > The paper lacks a clear definition of the Generation-Verification Gap, with no illustrative diagram and only complex, confusing formulas cluttered with symbols.
>
> We appreciate your suggestions on improving the presentation of the core concepts. In the revised pdf, we provide an illustration of the generation-verification gap (definition 2.1) in Figure.7, on top of page 17. Figure.7 uses rejection sampling and CoT-Binary verification as an example. In this case, the generation-verification gap can be understood as the accuracy difference between the generator and the filtered dataset, where the filtering criteria is determined by the verifier model. We keep it in the appendix for the revised submission to minimally impact the current main text - we will make sure to reorganize and include the diagram in the main text in the final version. We hope the illustrative diagram can help with understanding our key concept and definitions.
>
> > The authors fail to clearly outline the overall methodological framework or how the experimental process operates.
>
> Thank you for pointing this out. We hope our improvement on the presentation on the generation-verification gap in the last response can help clarify how the experiments operate. In addition, we add the following description in Section 3 (highlighted in blue):
> For each model $f$ and prompt $x$, we sample $128$ responses $y \sim f(x)$, and sample $1$ verification for each response, which defines the proxy utility score $\hat u_f(x,y)$. We mainly consider the rejection sampling setting (for completeness we investigate the RL setting in Section 5), and thus the weight function $w$ is the indicator function with either quantile or global threshold (c.r. Example 2.2). Then we calculate $\mathsf{gap}$ or $\mathsf{gap}_{\mathsf{rel}}$ according to Definitions 2.1 and 2.2, which is the accuracy difference between the filtered generations and the original generations.
>
> > The contributions to the community are relatively weak, as most of the conclusions are already known, such as the "verification gap scales monotonically with the model pre-training flops" and the "Saturation Limit and Cause of Saturation."
>
> We assume by "verification gap scales monotonically with the model pre-training flops" the reviewer refers to the result in Fig. 1, the scaling property of the relative generation-verification gap. Please let us know if this is not the case. We agree that many previous self-improvement papers have shown that a bigger model may have larger improvement, but there are a few reasons why the previous result is not sufficient.
> 1. Our paper defines the metric of generation-verification gap and we propose to decouple the gap and model update, while previous papers only measure the performance difference before and after the update (which combines the gap and model update), introducing confounders in the results.
> 2. The previous papers only measured two models or one model family at best, but our paper studies across multiple families of models. This enables us to discover a trend in terms of pretraining flops which has real practical implications and can serve as a guideline for future training.
> 3. Previous papers measure the absolute performance difference. While the performance differences might be monotone if only two or three models are measured, our large-scale study shows that this is not true - absolute gap does not scale monotonically with model size (c.r., Fig. 9 and Fig. 12 in the appendix). Our paper introduces the definition of relative gap which instead enjoys the scaling property and to our best knowledge, the concept of relative gap is not present in the previous literature. That said, we greatly appreciate any pointers on the relevant works to better contextualize our result!
>
> Regarding "Saturation Limit and Cause of Saturation", we agree that previous works have observed similar saturation phenomena in self-improvement [1,2], which we discussed in Section 5. The difference is, our study argues that one should disentangle the gap and the model update while studying iterative self-improvement. We highlight the discussion and add an additional remark in blue in the section paragraph of Section 5 (page 8). As for the “cause of saturation”, to our best knowledge we are not aware of any previous similar results at least in the context of self-improvement. However, we might have missed something as the field is rapidly growing and we are happy to add a discussion of any relevant work that the reviewer deems necessary.
>
> We hope our explanation highlights the contribution of our results and we will make sure to add more discussion in the paper.

---

> > ### Author Response · Authors · 2024-11-21
> > **Reponse (2/2)**
> >
> > > The experiments are primarily based on GSMK8k, which consists only of elementary-level math problems, making the results less convincing.
> >
> > Thank you for pointing this out. We are excited to share our new results on the more challenging MATH dataset [3], which consists of competition level math questions. We repeat the same experiments as in GSM8K on all model families, and an additional Qwen-2.5 model family that consists of 7 models. We observed similar results in both scaling experiments and iterative self-improvement experiments. We present the results in Appendix A in the revised submission.
> >
> > > In Section 3, the experimental setup has predetermined parameters for lm-evaluation-harness tasks. However, it is unclear why so many hyperparameters would affect the GV-Gap instead of being solely determined by differences between the verifier and the generator.
> >
> > Most of the hyperparameters are chosen from the default ones suggested in lm-evaluation-harness, and there are two major reasons for this:
> > - doing a hyperparam sweep on such a high-dimensional vector is infeasible given that a single run for one set of hyperparameters is already expensive.
> > - It helps to contextualize our results in the literature if we use the same hyperparameters as how these open models are evaluated.
> >
> > Also, note previous work in scaling law in downstream tasks [4] also only uses a fixed set up hyperparameters (except learning rate which is not applicable in our setting, and their learning rate is model-dependent). Nevertheless, to demonstrate the robustness of our findings, in Appendix B.1 in the revised submission, we record the results for Qwen-1.5 model family in GSM8K, with different sampling temperatures. We observe the similar results as long as the temperature is within a reasonable range.
> >
> > ### Reference
> > [1] Weizhe Yuan, Richard Yuanzhe Pang, Kyunghyun Cho, Sainbayar Sukhbaatar, Jing Xu, and Jason Weston. Self-rewarding language models. arXiv preprint arXiv:2401.10020, 2024.
> >
> > [2] Yiming Liang, Ge Zhang, Xingwei Qu, Tianyu Zheng, Jiawei Guo, Xinrun Du, Zhenzhu Yang,
> > Jiaheng Liu, Chenghua Lin, Lei Ma, et al. I-sheep: Self-alignment of llm from scratch through an
> > iterative self-enhancement paradigm. arXiv preprint arXiv:2408.08072, 2024.
> >
> > [3] Dan Hendrycks, Collin Burns, Saurav Kadavath, Akul Arora, Steven Basart, Eric Tang, Dawn Song, and Jacob Steinhardt. Measuring mathematical problem solving with the math dataset. arXiv preprint arXiv:2103.03874, 2021
> >
> > [4] Isik, Berivan, et al. "Scaling laws for downstream task performance of large language models." arXiv preprint arXiv:2402.04177 (2024).

---

> ### Comment · Reviewer_eq2C · 2024-12-01
> **Thanks for your rebuttal**
>
> Thanks for your response! The clarifications on Generation-Verification Gap diagrams, the details of the experimental procedures, and the additional resutl on the MATH dataset enrich the paper’s completeness. Most importantly, you clarified the concept of the relative discrepancy between the validator's verification accuracy and the generative model's inference accuracy—a point indeed rarely mentioned in previous studies.
>
> The concern regarding whether "Saturation Limit and Cause of Saturation" represents a new insight stems from the “Takeaway on iterative self-improvement” in original text. In fact, LLMs lose their ability to self-improve after 2 to 3 iterations due to repeated learning of the same data (oversampling 2 to 3 times), which prevents new knowledge from being introduced, or because new data is overridden by the original training data. This is conclusion widely known among LLM researchers.
>
> As a side note, the authors only used off-the-shelf offline LLMs for local inference. With the aid of vllm tools, even a 70B model takes less than an hour for most tasks in the lm-evaluation-harness, and only a few hours for the MMLU tasks. For a purely analytical paper, this isn't considered a high computational cost.
>
> Finally, I am pessimistic that this work will offer substantial help or guidance on how existing LLMs can self-improve their performance. Therefore, I am maintaining my current opinion and score.

---

> ### Author Response · Authors · 2024-12-01
> **Author Response**
>
> We thank the reviewer for their additional comments. We are glad to see that our rebuttal addresses all but one concern, which is the novelty regarding the "Saturation Limit and Cause of Saturation" takeaway.
>
> ### Saturation Limit and Cause of Saturation
>
> For the saturation limit result, as we have acknowledged in both orginal paper and in the previous rebuttal, similar phenomena have been observed in the preivous literature. We never intend to claim that we are the first to observe this phenomena, but we have argued in both the original paper and the rebuttal that previous studies have been using the wrong metric to study the saturation limit, and we propose to use the generation-verification gap to avoid confounders from the policy distrillation step. We refer the reviewer to the highlighted discussion in section 5, and we will make sure to make this point more clear by revising the takeaway in section 5.
>
> Regarding the cause of saturation, we would like to repeat our argument in the rebuttal that, to our best knowledge we are not aware of any previous similar results on examining potential cause of saturation, at least in the context of self-improvement. However, we might have missed something as the field is rapidly growing and we are happy to add a discussion of any relevant work that the reviewer deems necessary. To relate to the common knowledge about "no new knowledge is introduced during self-improvement", such a heuristic explaination might be plausible, but it does not offer insights on how to solve the problem. Our result on the effective diversity, on the other hand, offers a rigirous metric and practical implications on mitigating the saturation issue. We believe progresses in our field are often made through rigirous scientific studies.
>
> ### Computation
>
> Regarding the comment on the computation, we remark that our task is rather different from the reviewer's example. 1) for each question, we need to generate 128 responses, not 1 response. 2) we need to generate verifications for each generation. 3) tasks like MMLU only considers the logits of 4 tokens, while GSM8K and MATH requires generating hundreds or thousands of tokens for each generation. Thus the computation cost of our experiment is orders of magnitudes larger than the examples provided by the reviewer.
>
> ### Practical Guidance
>
> As a final remark, we want to remind the reviewer our results in addition to the iterative self-improvement results, and their practical implications:
> - the scaling result can help predict the self-improvement capability of bigger models and potentially lead to a way to find the compute optimal model with self-improvement.
> - our results on multiple tasks suggest the possibility of self-improvement on different types of downstreaning tasks.
> - the generalization of verification result suggests that the filtering threshold can transfer between models within the same task and verification mechanism.
> - the emsemble result suggests that in practice one can use multiple verification together to enhance self-improvement.
>
> We hope our response can clarify the reviewers concerns and change the reviewer's pessimistic perspective on the practical guidance our paper can provide, and we are happy to answer any additional questions.

---

> > ### Comment · Reviewer_eq2C · 2024-12-01
> > **Thank you very much for your responses**
> >
> > Thank you very much for your prompt and personalized responses!
> >
> > Through the Practical Guidance, I finally understand the value of this work, as described by the authors, in enhancing the self-improvement capabilities of LLMs.
> >
> > Nice work. I have increased my score from 5 to 6.

---

> > > ### Author Response · Authors · 2024-12-01
> > > **Author Response**
> > >
> > > We appreciate the reviewer's timely response and we are happy to see our response clarifies the reviewer's concerns. We will make sure to incorperate the discussions in the final version of the paper.

---

### Author Response · Authors · 2024-11-21
**General Response**

First we would like to thank all the reviewers for their constructive comments on new experiments. We believe the new results will greatly improve the paper.

In this thread we summarize the revision in the draft: we made a few presentational improvements in the main text on the introduction to the experiment protocols and we include the RL results in Figure 3. For the ease of presentation, we aggravate most of the additional results in Appendix A and B, detailed below. We will incorporate the new results into the main text in the final version. All new changes are highlighted with blue.


### MATH dataset (Appendix A)
As suggested by Reviewer eq2C and gkZY, GSM8K might not be a challenging enough benchmark. Thus we repeat our major experiments on the MATH dataset, a more challenging benchmark with competition level math questions. We record the details in Appendix A. In summary, we observe the same scaling phenomena on relative generation-verification gap with even stronger signals (more than one verification method elicits the scaling phenomena) and the same observations in iterative self-improvement (we have not completed the full iterative training loop for the bigger models and we will update the results once we completed the training).


### Different Sampling Temperature (Appendix B.1)
As suggested by Reviewer eq2C and 1Q5B, we conduct an ablation on different hyperparameter/decoding strategies. We tested sampling with temperature 1 and 0.5 and we observe the same scaling phenomena holds.


### Instruct Models (Appendix B.2)
For completeness, we include the scaling study for instruct models as suggested by Reviewer QqKc. We observe the results are more noisy due to the confounders introduced in the post-training process.


### Fixed Verifier (Appendix B.3)
To better understand the decrease in effective diversity, as suggested by Reviewer gkZY, we conduct iterative improvement with the gold label or a more powerful model as the verifier. Our result suggests that the decrease in effective diversity does not happen if we use the gold label as verifier, but it happens in the second setting. This suggests that one reason for this phenomenon is that the model concentrates on some wrong answers that the imperfect verifier labels as correct.


### Variance of the Verifier (Appendix B.4)
To better understand the noisiness of the verifier, as suggested by Reviewer QqKc, we measure the variance of the verifier in Appendix B.4. We observe that the verifier has near 0 variance on most of the generations.

---

> ### Author Response · Authors · 2024-11-27
> **Paper Update**
>
> We posted a revision of the paper that includes more experiment results, and we hope these results can further improve the completeness of our paper:
> - (Appendix A) For the result on MATH, we updated Figure 10 to include more rounds of self-improvement for the bigger models.
> - (Appendix B.2) For the result on instruct models, we added the results on the Llama-2-Chat model family.
> - (Appendix B.3) For the result on fixed verifier, we added one more round of finetuning and the result is updated in Figure 13.

---

### Meta-Review · Area_Chair_LZ8Q · 2024-12-13

**Metareview:**

Claims and findings:
The paper discusses self improvement in models of different stages of training.
They offer a way to evaluate the gap between generation and verification
With this way they show how this gap grows with more training, with better or worse model verifying or generating etc.

Strengths:
General framing yet convincing
Many experiments to support claims
Large potential for future improvement

**Additional Comments On Reviewer Discussion:**

It was vast, but summarized by the authors already.
Some concerns of the reviewers were raised leaving the paper in a good state.

---

### Decision · Program_Chairs · 2025-01-22

Accept (Oral)